



**Case-based formalization and reasoning method for**
**knowledge in digital terrain analysis ─ Illustrated by**
**determining the catchment area threshold for extracting**
**drainage networks**
**C.-Z. Qin[1,2,*] X.-W. Wu[1,3] J.-C. Jiang[4] A-X. Zhu[1,2,5,6]**
[1]{State Key Laboratory of Resources and Environmental Information System, Institute of
Geographic Sciences and Natural Resources Research, CAS, Beijing 100101, China}
[2]{Jiangsu Center for Collaborative Innovation in Geographical Information Resource
Development and Application, Nanjing 210023, China}
[3] {College of Resources and Environment, University of Chinese Academy of Sciences,
Beijing 100049, China}
[4] {Smart City Research Center, Hangzhou Dianzi University, Hangzhou, 310012, China}
[5] {Department of Geography, University of Wisconsin-Madison, Madison, WI 53706, USA}
[6] {Key Laboratory of Virtual Geographic Environment, Ministry of Education, Nanjing
210023, China}
[*]Correspondence to: C.-Z. Qin (qincz@lreis.ac.cn)
**Abstract**
Application of digital terrain analysis (DTA), which is typically a modeling process involving
workflow building, relies heavily on DTA domain knowledge of the match between the
algorithm (and its parameter settings) and the application context (including the target task,
the terrain in the study area, the DEM resolution, etc.), which is referred to as application-
context knowledge. However, existing DTA-assisted tools often cannot use application-
context knowledge because this type of DTA knowledge has not been formalized to be
available for inference in these tools. This situation makes the DTA workflow-building
process difficult for users, especially non-expert users. This paper proposes a case-based





formalization for DTA application-context knowledge and a corresponding case-based reasoning method. A case in this context consists of a series of indices that formalize the DTA application-context knowledge and the corresponding similarity calculation methods for case-based reasoning. A preliminary experiment to determine the catchment area threshold for extracting drainage networks has been conducted to evaluate the performance of the proposed method. In the experiment, 124 cases of drainage network extraction (50 for evaluation and 74 for reasoning) were prepared from peer-reviewed journal articles. Preliminary evaluation results show that the proposed case-based method is a suitable way to use DTA application-context knowledge to achieve a marked reduction in the modeling burden for users.

## 1 Introduction

Digital terrain analysis (DTA) is a useful approach because it can handle the complexity of GIS spatial analysis and has been widely used in geography and related fields (Wilson, 2012). More and more users, including many with little knowledge of DTA, are becoming involved in DTA applications. Use of DTA is typically a non-trivial workflow-building process consisting of organizing the various DTA tasks and specifying the algorithm (including parameter settings) for each task (Hengl and Reuter, 2009). This workflow-building process relies heavily on knowledge of the match between DTA algorithm specifications and the particular application context. However, current DTA-assisted tools (e.g., ArcGIS, GRASS, SAGA, White Box, TauDEM, etc.) provide very limited support during the DTA application modeling process (Qin et al., 2011). It is therefore difficult for users, especially those with little knowledge of DTA, to use DTA correctly and effectively.

Knowledge used during DTA workflow building can be classified into three types (Qin et al., 2011): 1) task knowledge, which describes the relationship between DTA tasks and their input/output; 2) algorithm knowledge, which is the meta-data of a DTA algorithm (including its parameters); and 3) the so-called application-context knowledge consisting of how to specify the suitable algorithm and its parameter settings for a DTA task according to the application context (such as application goals, study area characteristics, and DEM resolution) (Qin et al., 2013). This knowledge is called application-matching knowledge in Lu et al. (2012).

Among the three types of DTA knowledge, both task knowledge and algorithm knowledge have been formalized by means of rule or semantic networks (Russell and Norvig, 2009) and





hence can be used in existing DTA-assisted tools (e.g., ModelBuilder in ArcGIS). However,
application-context knowledge, which is crucial for building a suitable DTA model for a
specific application, is more difficult for a user to acquire than the other two types of
knowledge. Currently, there is no well-established formalization method by which DTA tools
can provide more effective assistance to DTA applications. This situation exists mainly
because this type of DTA knowledge is largely inaccurate and non-systematic, and often
exists only in documents for specific case studies (DTA application instances) or even just in
the experience of domain experts.
To solve this problem, this paper proposes a case-based formalization for DTA case studies
involving DTA application-context knowledge and a corresponding case-based reasoning
method. A DTA-assisted tool can then use this type of knowledge to reduce the difficulty of
DTA application modeling.

## 2   Basic idea

Cases are a commonly used way of formalizing non-systematic knowledge in artificial
intelligence. A case is a record of an existing problem-solving instance and its contextual
information, which has two requisite parts: the problem and the solution (Kaster et al., 2005).
The problem describes the application purpose of the case and its contextual information. The
solution is a set of methods (including their parameter settings) for achieving this purpose.
Note that the case is not the same as the concept of a prototype (Minda and Smith, 2001),
which can also use existing instances to describe empirical knowledge and has been applied in
the geographical domain (e.g., Qi et al., 2006; Qin et al., 2009). The prototype highlights the
representativeness of the instances, whereas the case does not. Currently, most DTA
application-context knowledge is empirical knowledge that often exists in application
instances and is difficult to formalize in as explicit rules or mathematical equations. In this
situation, the case is a suitable way to formalize DTA application-context knowledge (Lu et
al., 2012).
Case-based reasoning (CBR) (Schank, 1983) is a method of solving problems by referring the
solution of a new problem to the solutions of existing similar cases (Aamodt et al., 1994;
Watson and Marir, 1994). Compared with traditional rule-based knowledge representation
and reasoning methods, the case-based method can simplify knowledge acquisition into case
acquisition, with no need for an explicit expression model of domain knowledge (Watson and





Marir, 1994). Therefore, the case-based method is suitable for application domains that lack a
systematic expression of empirical domain knowledge. A case-based reasoning method could
be designed to use DTA application cases to reduce the difficulty of DTA application
modeling for users.
**3    Methodology**
According to the basic idea presented above, a case-based formalization methodology is
designed for DTA application instances containing application-context knowledge and the
corresponding inferences (Fig. 1). Case formalization and the corresponding case-based
reasoning method are the two main stages in the methodology.
**3.1    Case formalization**
Case formalization is the process of extracting and describing each individual case in a formal
way, so that the case can be retrieved by a corresponding case-based reasoning method.
Among the parts of a case, the case problem consists of a set of factors describing the
contextual information associated with the case. This set of factors is quantified using a set of
quantitative attributes that are directly involved in case-based reasoning. It is of crucial
importance to design and quantify these factors properly for case-based reasoning. The
solution part of a case, which records the candidate problem-solving result of the case-based
reasoning, is not necessary to participate in the reasoning procedure. The case output is an
optional part of the description that is used to record the status of factors describing the case
problem after the case occured (Kolodner, 1993). Therefore, the key to designing a case-based
formalization of DTA application-context knowledge is how to choose and quantify a set of
factors influencing DTA algorithm selection and parameter setting to describe the case
problem appropriately.
According to the characteristics of DTA application modeling, the case problem can be
described based on three groups of factors that influence DTA algorithm selection and
parameter setting (Table 1): application purpose, data characteristics, and study area
characteristics. For example, a single flow-direction algorithm (e.g., the classic D8 algorithm)
is suitable for deriving flow accumulation from a SRTM DEM (with a resolution of 90 m) for
drainage network extraction in high-relief areas, whereas a multiple flow-direction algorithm
should be used with a 10-m DEM created from a contour map for estimating detailed spatial





distribution of flow accumulation and other related regional topographic attributes (such as
topographic wetness index) in a low-relief area. In this example, the choice between a single
flow-direction algorithm and a multiple flow-direction algorithm is influenced by the
application purpose (i.e., the DTA task of drainage network extraction or deriving the spatial
distribution of regional topographic attributes), data characteristics (i.e., a SRTM DEM with
90-m resolution or a contour-originated DEM with fine resolution), and study area
characteristics (mainly terrain condition, e.g., high or low relief). This example shows the
typical content of application-context knowledge in DTA application modeling.
Among these three groups of factors, the application purpose can be formalized by an
enumeration-type variable. Data characteristics can be mainly described by the spatial
resolution of the DEM, the type of data source, etc. In particular, the spatial resolution, which
is often indicated by the grid cell size for the widely used grid-based DTA, is the most
important factor among the data characteristics. The group of factors describing the study area
characteristics related to DTA application-context knowledge could include location, area,
terrain condition, and other environmental conditions (such as climate, geology, etc.).
Generally, terrain condition in a study area comprehensively reflects the influence of all
geographical processes on the landforms in the area. This means that terrain condition might
be one of the most important factors influencing the DTA algorithm selection and parameter
settings. Because of its comprehensiveness, the terrain condition factor should be quantified
by multiple attributes during case-based formalization of DTA application-context knowledge.
Different designs of the quantitative attributes will result in different case-based methods.
In a case-based formalization of DTA application-context knowledge, the solution part of a
case can be formalized by recording the name of the DTA algorithm and the corresponding
parameter values used in this case, which is much simpler than describing the case problem.
The optional output part of the case-based formalization does not currently need to be
considered for the DTA domain because normally there is no change in the application
context of a DTA application case when the DTA model is applied.

## 3.2 Case-based reasoning method

Case-based reasoning is based on the principle that solutions for similar problems are often
similar, even identical. Therefore, a new DTA application problem can be formalized in the
same way as the case problem part in a prepared DTA case base and then be used in case-



based reasoning by calculating the similarity between this new application problem and the
problem part of each case in the case base. The solution of the case with the highest similarity
is reused for the new DTA application problem. Note that in the conceptual framework of a
case-based reasoning method, the solution of the retrieved case with the highest similarity
might be further revised to adapt to the new application problem when the final solution for
the new application problem is retained in the case base (Watson and Marir, 1994). However,
the method developed in this preliminary study currently considers neither the revision nor
the retention process.
Calculating the similarity between a new DTA application problem in case format and the
problem part of each case in the DTA case base consists of the following two steps:
Step 1. Calculate the similarity of each individual attribute between the new application
problem and the problem description of an existing case. As usual the range of the similarity
value is [0, 1]; the larger the value, the more similar are the two cases. As mentioned above,
the attributes used to formalize the problem part of a DTA application case may have different
value types, such as enumeration type (e.g., application purpose), single-value type (e.g.,
spatial resolution and area), or even a frequency distribution (e.g., hypsometric curve). For
each attribute, a similarity function should be designed correspondingly to quantify the
deviation on this attribute between the new application problem and an existing case. The
design is generated in an empirical way and should match the domain knowledge.
Step 2. Synthesize the similarity values for every individual attribute to calculate the overall
similarity between the new application problem and the problem description of an existing
case. In the geographical domain, a minimum operator based on the limiting factor principle
is often used to synthesize similarity values on multiple attributes (Qin et al., 2009).

## 4   Design of a detailed method

In this section, the methodology presented in the previous section is concretized by designing
a detailed case-based formalization method for DTA application instances containing
application-context knowledge and the corresponding inferences. The key issue in method
design is designing a set of quantitative attributes describing the case problem and the
similarity function on each individual attribute. Because the gridded DEM is widely used in





practical applications, this method is designed mainly for grid-based DTA, although the
methodology is available for both grid- and vector-based DTA.
**4.1    Selection of attributes**
The set of quantitative attributes should be designed to effectively reflect the contextual
information related to DTA application modeling, and be fit for the case-based reasoning to
follow. The purpose of a DTA application case is naturally described by an enumeration-type
attribute, i.e., the name of the target task. Here, cell size has been chosen as the attribute to
quantify the data characteristics of a DTA application case; other potential factors (such as
type of data source) for describing data characteristics are not currently considered.
To describe the study area characteristics of a DTA application case, the area and the terrain
condition of the case are considered in the current method. Like cell size, area is an attribute
with a single numeric value. Terrain condition is an important and comprehensive factor
indicating the difference in study area characteristics between a new DTA application
problem and an existing case.
In this study, the three following aspects were designed to describe the terrain condition factor
empirically:
1) Relief. The relief attribute is a commonly used value to describe the overall terrain
condition of a study area, whether it is steep or gently sloping.
2) Slope distribution. The slope distribution provides information on the proportions of
different intensities of local relief in the area, which cannot be described by the relief in the
overall area and is useful for judging the reasonableness of a DTA algorithm selection and its
parameter settings. To describe in detail the slope distribution in a study area, we quantified it
by a relief-slope frequency distribution. For this purpose, the slope gradient was divided into
seven grades: 0°–3°, 3°–8°, 8°–15°, 15°–25°, 25°–35°, 35°–45°, and 45°–90° (Tang et al.,
2006). The relief of the study area was classified into one of ten levels with equal step. The
relief-slope frequency distribution obtained in this way is a two-dimensional table with 10
level ×7 grade data items. Considering the influence of DEM resolution on the slope gradient
calculation (Chang et al., 1991; Grohmann, 2015), a relief-slope cumulative frequency
distribution were used here instead of the relief-slope frequency distribution to provide a
quantitative description that relieves the DEM resolution effect. The relief-slope cumulative
frequency in each relief level is calculated by accumulating the number of cells within each



slope gradient grade from low to high grade in this relief level. Note that the 10-level division
of elevation considers only the relative relationship among the elevation levels inside the
study area. The elevation level might consist of a distinct elevation step for a study area, in
which case the relief of the study area would be ignored for this attribute. This proposed
design appears to be not only a convenient way to automate similarity calculations in case-
based reasoning, but also reasonable because the relief attribute reflects the relief information
throughout the study area.
3) Landscape development stage for the study area, which can provide information on the
geomorphic processes (mainly hydrological erosion process) affecting terrain conditions in a
study area (often a watershed). This information is useful for judging the reasonableness of a
choice of DTA algorithm and its parameter settings related to hydrological and erosion
processes. In this study, the hypsometric curve (Strahler, 1952), which is normally used to
analyze the landscape development stage of river basins, was used as an attribute to describe
this aspect.
In the proposed method, location is not used as a study area characteristics. This decision was
made because the influence of the study area location in DTA application-context knowledge
could be reflected by the terrain condition of the study area, which directly impacts the choice
of DTA algorithm and parameter settings and has already been considered in the method. For
similar reasons and for the sake of brevity, in the proposed method, environmental conditions
other than terrain condition are not considered.
Table 2 lists the attributes used to formalize a case problem in this method.

### 4.2   Similarity function on each individual attribute

The design of the similarity function for an individual attribute should be compatible with the
value type of the attribute and in accord with domain knowledge regarding the level of
similarity due to the difference in the attribute value between the new application problem and
an existing case. For an attribute of the enumeration type, its similarity value between a new
application problem and an existing case can be calculated by a Boolean function (Fig. 2a).
When the attribute values are matched, the similarity value is 1, otherwise it is 0.



For an attribute of the single numeric value type, two commonly used kinds of basic similarity
function are considered in this study: the linear function and the bell-shaped function (Fig. 2).
Both kinds of similarity function accord with common sense in that the similarity is 1 for the
minimum difference (i.e., zero) of attribute value, and the greater the difference in attribute
value, the lower is the similarity. With the linear function, the similarity value is set to 0 or 1
when the absolute difference of the attribute between a new application problem and an
existing case reaches its maximum or minimum value. The similarity can be calculated for
other difference values by linear interpolation (Fig. 2b). The similarity function based on a
linear function fits the specification that the maximum difference in attribute values can be
preset.
With the bell-shaped function, the maximum difference in attribute values is not easy to
preset and does not need to be. A simplified version of the commonly used bell-shaped
function (Shi et al., 2005; Qin et al., 2009; Fig. 2c) is:
$S = e^{(|v_{new} - v_{case}|/w)^{0.5} \ln(0.5)}$.                                    (1)
where $S$ is the similarity between a new application problem and an existing case;
$v_{new}$ and $v_{case}$ are attribute values of the new application problem and the existing case
respectively; and $w$ is the shape-adjusting parameter of the function. When the difference
between $v_{new}$ and $v_{case}$ is equal to $w$, the similarity $S = 0.5$ (Fig. 2). Some sort of numerical
transformation on the attribute value could be necessary for the similarity calculation to yield
a reasonable reflection of the similarity level due to differences in the attribute.
For an attribute of more complex type (such as a frequency distribution), a quantitative index
should be designed to quantify the difference in an attribute between a new application
problem and an existing case. Then the similarity on this attribute can be calculated based on
this index, similarly to the single numeric-value type.
Based on these kinds of basic similarity function, similarity functions for each individual
attribute used for case-based reasoning in this paper were designed as shown in Table 2. The
following discussion introduces them one by one.
**4.2.1  Name of target task**
The name of the target task is an attribute of the enumeration type. The similarity value for
this attribute between a new application problem and an existing case can be calculated by a




Boolean function. When the names of two target tasks match, the similarity value is 1,
otherwise it is 0.

### 4.2.2  Cell size

Note that the difference in magnitude of cell size can better reflect the level of similarity
between DTA applications than the numerical difference in cell size. The greater the
difference in magnitude, the lower is the similarity. According to this knowledge, a base-10
logarithmic transformation was applied to the cell size during the similarity calculations.
Because it is not easy to preset the maximum of the attribute value after logarithmic
transformation, the bell-shaped function based on Eq. (1) was used to calculate similarity for
cell size. Furthermore, $w$ in Eq. (1) is set to 0.5, which means that the similarity in cell size
between a new application problem and an existing case will decrease to 0.5 when their
difference in cell size reaches one order of magnitude (e.g., 1 m vs. 10 m, or vice versa). The
similarity function used in the proposed method for cell size is shown in Table 2.

### 4.2.3  Area

Like cell size, area is also an attribute of the single numeric value type. The greater the
difference in magnitude between two areas, the lower is their similarity on area. Similarly to
the design for the cell size attribute, a base-10 logarithmic transformation is applied to the
area attribute and then the similarity function for this attribute is designed based on the bell-
shaped function. The $w$ in Eq. (1) has been set to 1.5 for the area attribute by trial and error
(see Table 2).

### 4.2.4  Relief

The greater the difference in relief value between a new application problem and an existing
case, the lower is the similarity. The maximum difference in relief values between two DTA
application areas can be preset due to the geometric nature of the Earth. Hence, the similarity
function for the relief attribute was designed as a linear function using the absolute difference
between the relief of the new DTA application problem and that of existing case.
Corresponding to a zero similarity value, the maximum difference between two relief values
is the larger of the relief differences between the new application problem values and each of
two extreme cases (a flat area with zero relief, and an area with relief from the 8848 m of





Mount Everest to sea level). The similarity function used in this method for the relief attribute
is shown in Table 2.

### 4.2.5 Relief-slope cumulative frequency distribution (describing the slope distribution)

The relief-slope cumulative frequency distribution is a two-dimensional table with 10 level ×
7 grade data items. This two-dimensional table can be viewed as a DEM having a volume
with a constant projected area. The greater the overlap in volume between the distribution of a
new application problem and that of an existing case, the higher is the similarity. Therefore,
the similarity function for the relief-slope cumulative frequency distribution was designed as
the ratio of the intersection volume to the union volume between two distributions (Table 2).

### 4.2.6 Hypsometric curve (describing the landscape development stage)

The hypsometric curve is often summarized as a single numeric value, the hypsometric
integral (HI, with a value range of [0,1]), which can be used to classify landscape
development into three stages: youth (HI > 0.6), maturity (0.35 < HI < 0.6), and old age (HI <
0.35) (Strahler, 1952). The HI was used to design a similarity function for the hypsometric
curve between a new application problem and an existing case, which is a linear function
using the absolute difference of their HI values. When the absolute difference in HI is 0, the
corresponding similarity is 1. The similarity is 0 for the maximum possible deviation from the
HI of the new application problem (see Table 2).
The overall similarity between a new application problem and an existing case is calculated as
the minimum of all similarity values for every individual attribute between the new
application problem and the existing case.

## 5   Experiment

### 5.1   Experimental design

The extraction of a drainage network, one of the most important DTA applications, was taken
as an example to evaluate the proposed method. The general workflow of river network
extraction based on a gridded DEM includes the following three DTA tasks in sequence: 1)



preparing a DEM by filling in the artificial pits and removing absolutely flat areas; 2) using a
flow direction algorithm to derive the spatial distribution of the catchment area (CA); and 3)
setting a CA threshold to extract the drainage network from the spatial distribution of the CA.
In this DTA workflow, proper selection of the DTA algorithms (such as the DEM preparation
algorithm and the flow direction algorithm) and of parameter values (e.g., the CA threshold)
is based on DTA application-context matching knowledge. In many geographical information
systems (such as ArcGIS), the DTA algorithm used for drainage network extraction has often
been set to a default selection (e.g., the D8 algorithm as the default flow direction algorithm)
in such a way that the user cannot choose the DTA algorithm. The CA threshold is an
empirical parameter which varies with the study area characteristics and affects the extraction
results directly. Current DTA-related tools often leave the choice of CA threshold for
drainage network extraction to the user. However, it is difficult for users, especially non-
expert users, to determine the appropriate threshold for their applications.
Therefore, this experiment was designed to focus on using the proposed method to determine
the CA threshold for drainage network extraction. This means that the cases used in this
experiment have the same name as the target task, i.e., drainage network extraction. The core
of the solution part of the cases is the parameter value, i.e., the CA threshold. Although this
experiment is somewhat simplified, we believe that it can evaluate the proposed method as
effectively as an experiment with a more complex design.

### 5.1.1 Preparation of a case base

The case base prepared for this experiment includes 124 cases of drainage network extraction
(Fig. 3). Each case originated from an article related to the target task that was recently
published in mainstream journals of related domains (such as Water Resources Research,
Hydrology and Earth System Sciences, Hydrological Processes, Computers & Geosciences,
Advances in Water Resources; see the Appendix document for the list of the articles used for
cases). These articles are supposed to provide good solutions for their specific study areas
based on experts' experience and knowledge of the target task.
Each case was manually prepared from a journal article. The main work involved in preparing
the case problem was extracting each attribute of the study area, whereas the work involved in
preparing the case solution consisted of extracting the CA threshold used in the article.
Normally, the cell size used is clearly stated in the article and can be filled in as the





corresponding case attribute. However, this is often not true for other attributes. Therefore, an
automatic program was applied to a free DEM dataset of the study area (mainly an SRTM
DEM with a resolution of 90 m and an ASTER GDEM with a resolution of 30 m) to derive
the other attributes (such as area, relief, relief-slope cumulative frequency distribution, and
hypsometric curve) for each case. For the solution part of each case, the CA threshold given
explicitly in each article was recorded directly. If the CA threshold was shown only implicitly
in the drainage network figure in an article, it was determined based on visual comparison
between the drainage network given in the article and those extracted from the DEMs used to
prepare other attributes of this case, using trial and error.
**5.1.2 Evaluation method**
Among the 124 cases in the case base, 50 cases randomly selected were used as independent
evaluation cases, which were assumed to be new application problems without a solution and
were solved by the reasoning method proposed. The other 74 cases were set aside as the case
base to be used by the proposed case-based reasoning method.
To perform a quantitative evaluation of the results from the proposed method on the 50
evaluation cases, an index was used, specifically the relative error of river density ($E$):
$$E = \frac{|RiverDensity^{origin} - RiverDensity^{reason}|}{RiverDensity^{origin}}. \tag{2}$$
where $RiverDensity^{origin}$ and $RiverDensity^{reason}$ are the river density values of a new
application problem (i.e., an evaluation case), obtained respectively from the original CA
threshold and the CA threshold solution obtained from the 74-case base by the proposed
reasoning method. $E$ is the relative error in river density for the evaluation case. The smaller
the value of $E$, the more reasonable is the result obtained for the evaluation case using the
proposed method. Four levels of $E$ were established empirically to reflect the reasonableness
level: reasonable ($E \in [0,0.1]$), acceptable ($E \in (0.1,0.25]$), questionable ($E \in (0.25,0.5]$), and
unreasonable ($E \in (0.5,+\infty)$). Representative cases were also selected to discuss the
reasonableness of its similarity result obtained using the proposed method. The relationship
between $E$ and the similarity value of the solution case to the evaluation case was also
analyzed to discuss the performance of the proposed method.





## 5.2  Experimental results and discussion
Table 3 lists the results of 50 evaluation cases solved by the proposed method using the case
base presented in the previous section. The similarities between every evaluation case and its
most similar case as reasoned by the proposed method were found in this experiment to lie
within a value range from 0.47 to 0.9. The higher the similarity, the lower is the uncertainty of
the result from the proposed method.
According to the relative error of river density (E), the counts of evaluation cases with
reasonable, acceptable, questionable, and unreasonable results are 26, 16, 3, and 5
respectively (Table 3). This shows that the proposed method performs satisfactorily. Taking
the results on two evaluation cases, Godavari [1053] (the "[1053]" means that the original CA
threshold recorded in the Godavari case was 1053 km$^2$) and Burdekin [502] ("[502]" defined
similarly) as examples, their most similar cases in the case base as reasoned by the proposed
method were KrishnaRiver [908.08] and MahanadiRiver [891] respectively. The CA
threshold values from the solution of the most similar cases (908.08 km$^2$ and 891 km$^2$) were
applied respectively to the Godavari and Burdekin evaluation cases. The extracted drainage
networks are with close spatial distribution as those extracted with the original CA thresholds
of the evaluation cases (Fig. 4). Their values of relative error of river density are 0.07
(reasonable level) and 0.24 (acceptable level) respectively.
The evaluation results with questionable and unreasonable levels also have lower similarities.
This means that there is no case in the current case base that has an application context highly
similar to that of the evaluation case. Hence, the solution from the proposed method has
higher uncertainty and might lead to questionable or even unreasonable application results for
new application problems. Taking the result for the YbbsRiver [1.01] evaluation case (E=0.4;
questionable) as an example, the similarities between this evaluation case and other cases in
the case base depend mostly on the similarities on the cell size attribute during the case-based
reasoning process proposed in this paper (Table 4). Because the cell size of the YbbsRiver
case is 10 m, which is relatively unlike cell size (30 m or 90 m) of most other cases in the case
base, the overall similarities between this evaluation case and these cases in the case base are
mainly limited by the individual similarity on cell size when synthesizing the similarities on
individual attributes by the proposed method. Furthermore, Table 4 shows that the CA
threshold values of the cases with the top 10 highest similarity values to the YbbsRiver
evaluation case would make the E value of the application result for the evaluation case



questionable or even unreasonable ($E$: 0.33–21.73). The solution selected by the proposed
method achieved a relatively better application result.
As for the reasoning results on the Kasilian [0.08] evaluation case ($E$=0.63; unreasonable)
using the proposed method, no individual attribute has a controlling effect on the overall
similarity between the Kasilian evaluation case and the other cases in the case base (Table 5).
The CA threshold values of the cases with the top 10 highest similarity values to the Kasilian
evaluation case would almost always lead to an unreasonable $E$ value of the application result
for the evaluation case ($E$: 0.48–0.92). The similarities between this evaluation case and the
cases in the case base are lower (Table 5). This problem could be mitigated by extending the
case base to contain cases with more combinations of data characteristics and study area
characteristics.
The distribution of the similarity results of the evaluation cases from the proposed method
among the reasonableness levels of the drainage network results using the solved CA
thresholds was also analyzed (Table 6). All solution cases with higher similarity (above 0.7)
to the evaluation cases produced reasonable and acceptable drainage network results, whereas
solution cases with lower similarity (below 0.7) often produced the questionable and
unreasonable drainage network results. This shows the effectiveness with which similarity
reflects uncertainty in the proposed method.

## 20 6  Summary

Although DTA application-context knowledge is of key importance in building an appropriate
DTA application, currently this type of knowledge has not been formalized to be available for
DTA-assisted tools to relieve the modeling burden of DTA users (especially non-expert users).
This paper has proposed a case-based methodology for formalizing DTA application-context
knowledge and corresponding case-based reasoning. A detailed method based on this
methodology has been developed. Taking drainage network extraction from a gridded DEM
as an application example, 124 cases (50 for evaluation and 74 for reasoning) of drainage
network extraction from peer-reviewed journal articles were used to evaluate the performance
of the proposed method. Preliminary evaluation results show the reasonableness of the
proposed case-based method.





Additional research is needed to enhance the proposed method. Currently the proposed
methodology is implemented as a primary method in this paper. The design for the individual
attributes and their quantification in each case could be improved to describe the application-
context knowledge in a more adaptive way for various DTA application targets. Another
possible improvement to the method would be to revise the solution part of the case as
suggested by case-based reasoning before applying the solution to the new application
problem. The possibility of synthesizing the solutions of the cases in the base with higher
similarity to build a solution to the new application problem could be also explored.
Automatic or semi-automatic methods of creating cases are needed to speed up the expansion
of the case base (not only for the current target task, but also for other DTA application tasks).
An expanded case base containing as many cases as possible with more combinations of all
kinds of characteristics would improve the application effectiveness of the proposed method.
The size of the case base also matters when evaluating the effectiveness of the case-based
reasoning method and its successive versions. However, current cases used in the experiment
were mainly manually prepared from journal articles, except for certain attribute calculations
(e.g., relief, hypsometric curve), for which an automatic computer program was used. This
inefficient way of preparing cases needs to be improved through automatic or semi-automatic
case-extraction methods.
**Acknowledgements**
This study was supported by the National Natural Science Foundation of China (No.
41422109, 41431177), and the National Science & Technology Pillar Program of China (No.
2013BAC08B03-4).

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



1    Table 1. General composition of DTA application-context knowledge in a case-based

2    formalization.

| Part of case | Composition of DTA application-context knowledge |
|---|---|
| Case problem | Application purpose |
| | Data characteristics (spatial resolution, data source, etc.) |
| | Study area characteristics (location, area, terrain condition, other environmental conditions) |
| Case solution | DTA algorithm used and its parameter settings |
| Case output (optional) | (not considered in the current DTA application) |





Table 2. Attributes used in this study to formalize the case problem and the corresponding
similarity functions for case-based reasoning using DTA application-context knowledge.

| DTA application context | | | Similarity function |
|---|---|---|---|
| Factor group | Factor | Attribute | |
| Application purpose | Target task type | Name of target task | Boolean function |
| Data characteristics | Spatial resolution | Cell size (m) | $S_i = 2^{-(2\|lgR_{new} - lgR_i\|)^{0.5}}$ |
| Characteristics of study area | Area | Area (km$^2$) | $S_i = 2^{-(\|lgArea_{new} - lgArea_i\|/1.5)^{0.5}}$ |
| | | Relief (m) | $S_i = 1 - S_i^{'}/max(8848 - Relief_{new}, Relief_{new})$ $S_i^{'} = \|Relief_{new} - Relief_i\|$ |
| | Terrain condition | Relief-slope cumulative frequency distribution (describing slope distribution) | $S_i = \dfrac{Intersect(RlfSlp_{new}, RlfSlp_i)}{Union(RlfSlp_{new}, RlfSlp_i)}$ |
| | | Hypsometric curve (quantifying the landscape development stage) | $S_i = 1 - S_i^{'}/max(1 - HI_{new}, HI_{new})$ $S_i^{'} = \|HI_{new} - HI_i\|$ |

Note: $S_i$ is the similarity (value range: [0, 1]) of an individual attribute between a new
application problem and the $i$-th case; $R_{new}$, $R_i$ are the DEM resolutions (m) of the new
application problem and the $i$-th case respectively; $Area_{new}$, $Area_i$ are the areas (km$^2$) of the
new application problem and the $i$-th case respectively; $Relief_{new}$, $Relief_i$ are the relief (m)
of the new application problem and the $i$-th case respectively; $RlfSlp_{new}$, $RlfSlp_i$ are the
histograms of the relief-slope cumulative frequency distributions of the new application
problem and the $i$-th case respectively; and $HI_{new}$, $HI_i$ are the hypsometric integrals of the
new application problem and the $i$-th case respectively.



1     Table 3. Evaluation results of the proposed method.

| Evaluation case [original CA threshold (km$^2$)] | Most similar case [CA threshold (km$^2$)] | Similarity | $E$ | Reasonableness level |
|---|---|---|---|---|
| UpperRhone [81] | KernRiver [81] | 0.83 | 0 | |
| MicaCreek1 [0.03] | MicaCreek2 [0.03] | 0.85 | 0 | |
| WillowRiver [40.5] | Bowron [40.5] | 0.89 | 0 | |
| YamzhogYumCo [12.15] | CedoCaka [12.15] | 0.75 | 0 | |
| Stanley [0.2] | Pettit [0.2] | 0.73 | 0 | |
| Alturas [0.2] | Pettit [0.2] | 0.68 | 0 | |
| WarregoSC2 [4.42] | WarregoSC4 [4.33] | 0.83 | 0.01 | |
| Toachi [3.13] | SanPabloLaMana [3.07] | 0.76 | 0.01 | |
| FuRiver [0.009] | CameronHighlands [0.0093] | 0.64 | 0.02 | |
| Davidson [0.48] | UpperMcKenzie [0.5] | 0.59 | 0.02 | |
| Komati [36.64] | Bowron [40.5] | 0.60 | 0.04 | |
| UpperTaninim [0.52] | Bellever [0.59] | 0.81 | 0.05 | |
| Crocodile [36.30] | Bowron [40.5] | 0.74 | 0.05 | Reasonable |
| Cheakamus [8.1] | LiWuRiver [9] | 0.80 | 0.05 | |
| Susquehanna [810] | DoloresR_Cisco [763.17] | 0.71 | 0.05 | |
| RoudbachPlaten [0.32] | HJA [0.27] | 0.80 | 0.06 | |
| Godavari [1053] | KrishnaRiver [908.08] | 0.80 | 0.07 | |
| Gard [8.09] | JuniataRiver [6.98] | 0.69 | 0.07 | |
| Urola [5.22] | OitaRiver [6.48] | 0.79 | 0.07 | |
| UpperDalya [0.45] | Bellever [0.59] | 0.82 | 0.08 | |
| WarregoSC3 [5.05] | WarregoSC4 [4.33] | 0.77 | 0.08 | |
| SanJuanR_Bluff [708.35] | ColoradoR_Cameron [794] | 0.87 | 0.08 | |
| Monastir [3.47] | Baba [4.19] | 0.80 | 0.08 | |
| SouthPark [24.3] | CooperRiver [29.34] | 0.78 | 0.09 | |
| Rhone [398.97] | PoRiver [486] | 0.86 | 0.1 | |
| Bishop_Hull [0.86] | Brue [0.70] | 0.78 | 0.1 | |





| | | | | |
|---|---|---|---|---|
| AlzetteEttel [0.23] | Bellebeek [0.31] | 0.76 | 0.12 | |
| PedlerCreek [0.41] | Bellever [0.59] | 0.70 | 0.12 | |
| Fengman [243] | UpperGuadiana [324] | 0.66 | 0.14 | |
| Cauvery [1053] | ColoradoR_Cameron [794] | 0.77 | 0.15 | |
| MiddleColorado [5.93] | WarregoSC4 [4.33] | 0.85 | 0.15 | |
| LuckyHills [6.3] | SouthForkNew [2.7] | 0.71 | 0.15 | |
| Limpopo [987.22] | DoloresR_Cisco [763.17] | 0.61 | 0.16 | |
| LittlePiney [2.84] | Blackwater [4.35] | 0.86 | 0.17 | Acceptable |
| ChiJiaWang [0.34] | ErhWu [0.23] | 0.80 | 0.17 | |
| Hailogou [2.03] | SanPabloLaMana [3.07] | 0.68 | 0.18 | |
| Batchawana [0.75] | ClearCreek [1.22] | 0.58 | 0.2 | |
| Liene [5.37] | LiWuRiver [9] | 0.74 | 0.2 | |
| Zwalm [0.36] | Haean [0.55] | 0.73 | 0.2 | |
| TapajosRiver [2720] | SaoFrancisco [5160] | 0.67 | 0.23 | |
| Burdekin [502] | MahanadiRiver [891] | 0.90 | 0.24 | |
| Garonne [247.68] | PoRiver [486] | 0.71 | 0.24 | |
| NorthEsk [1.22] | SanPabloLaMana [3.07] | 0.63 | 0.33 | |
| YbbsRiver [1.01] | Davidson [0.48] | 0.69 | 0.43 | Questionable |
| Cordevole [0.68] | SouthForkNew [2.7] | 0.69 | 0.46 | |
| NarayaniRiver [130] | Durance [51.21] | 0.51 | 0.52 | |
| YaluTsangpo [81.56] | SalmonRiver [486] | 0.47 | 0.55 | |
| Kasilian [0.08] | Haean [0.55] | 0.63 | 0.63 | Unreasonable |
| UpstreamGarza [0.2] | NorsmindeFjord [4.05] | 0.69 | 0.74 | |
| Zhanghe [33.11] | Lonquen [7.29] | 0.69 | 1.06 | |



1    Table 4. Top 10 similarity values between the YbbsRiver evaluation case and existing cases

2    as reasoned by the proposed method.

| Case name | Similarity value on individual attribute | | | | | Overall similarity | E |
|---|---|---|---|---|---|---|---|
| | Cell size | Area | Relief | Relief-slope distribution | Hypsometric curve | | |
| UpperMcKenzie | 1 | 0.73 | 0.90 | 0.62 | 0.92 | 0.62 | 0.4 |
| XianNanGou | 0.58 | 0.61 | 0.88 | 0.59 | 0.76 | 0.58 | 21.73 |
| NorsmindeFjord | 0.58 | 0.74 | 0.84 | 0.64 | 0.91 | 0.58 | 0.44 |
| Pettit | 1 | 0.56 | 0.96 | 0.62 | 0.76 | 0.56 | 1.19 |
| Bellebeek | 0.54 | 0.69 | 0.83 | 0.54 | 0.81 | 0.54 | 0.73 |
| Haean | 0.51 | 0.65 | 0.94 | 0.78 | 0.93 | 0.51 | 0.33 |
| MicaCreek2 | 0.51 | 0.53 | 0.89 | 0.62 | 0.75 | 0.51 | 5.23 |
| SouthForkNew | 0.51 | 0.69 | 0.89 | 0.76 | 0.52 | 0.51 | 0.35 |
| Babaohe | 0.51 | 0.57 | 0.88 | 0.73 | 0.90 | 0.51 | 0.73 |
| ClintonRiver | 0.51 | 0.59 | 0.85 | 0.56 | 0.55 | 0.51 | 0.79 |





1    Table 5. Top 10 similarity values between the Kasilian evaluation case and existing cases as

2    reasoned by the proposed method.

| Case name | Similarity value on individual attribute | | | | | Overall similarity | E |
|---|---|---|---|---|---|---|---|
| | Cell size | Area | Relief | Relief-slope distribution | Hypso metric curve | | |
| Haean | 0.63 | 0.92 | 0.83 | 0.83 | 0.93 | 0.63 | 0.63 |
| SanPabloLaMana | 0.61 | 0.61 | 0.74 | 0.60 | 0.76 | 0.60 | 0.84 |
| Brue | 0.61 | 0.67 | 0.73 | 0.59 | 0.88 | 0.59 | 0.66 |
| OitaRiver | 0.61 | 0.57 | 0.95 | 0.73 | 0.96 | 0.57 | 0.91 |
| Baba | 0.61 | 0.55 | 0.98 | 0.83 | 0.97 | 0.55 | 0.87 |
| JuniataRiver | 0.63 | 0.55 | 0.78 | 0.64 | 0.86 | 0.55 | 0.92 |
| NorsmindeFjord | 0.54 | 0.74 | 0.71 | 0.72 | 0.95 | 0.54 | 0.87 |
| Lonquen | 0.61 | 0.52 | 0.82 | 0.73 | 0.93 | 0.52 | 0.92 |
| HJA | 0.63 | 0.90 | 0.86 | 0.51 | 0.64 | 0.51 | 0.48 |
| Bellever | 0.61 | 0.78 | 0.74 | 0.50 | 0.68 | 0.50 | 0.63 |



Table 6. Relationship between $E$ and the similarity value of the solution case to the evaluation
case.

| | $S \in [0.8,1]$ | $S \in [0.7,0.8)$ | $S \in [0.6,0.7)$ | $S \in [0,0.6)$ |
|---|---|---|---|---|
| $E \in [0,0.1]$ | 10 | 11 | 3 | 2 |
| $E \in (0.1,0.25]$ | 3 | 8 | 4 | 1 |
| $E \in (0.25,0.5]$ | 0 | 0 | 3 | 0 |
| $E \in (0.5,+\infty)$ | 0 | 0 | 3 | 2 |





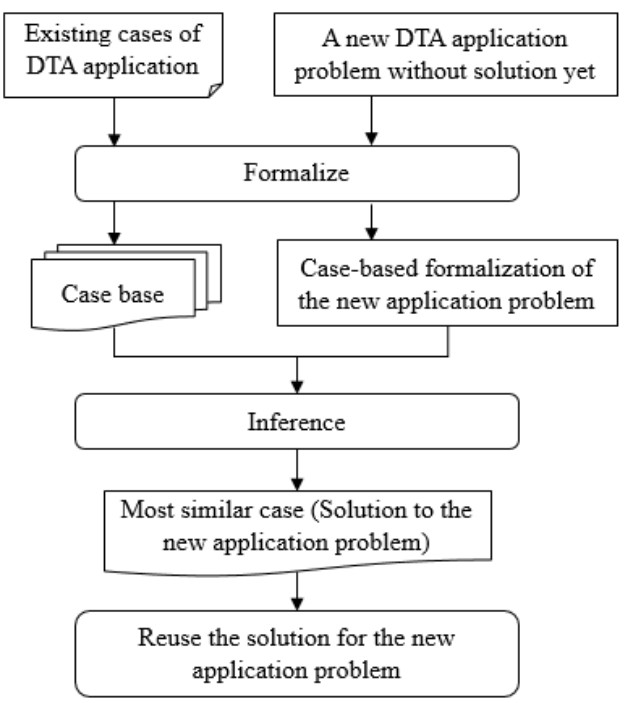

Figure 1. Structure of the case-based formalization and reasoning method for DTA
application-context knowledge.





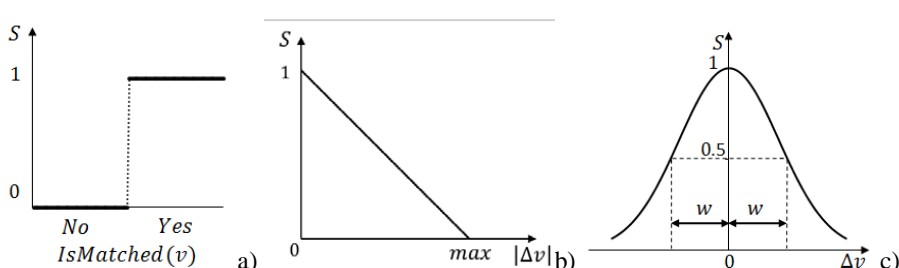

3    Figure 2. Basic kinds of similarity function: a) Boolean function; b) linear function; c) bell-

4    shaped function.





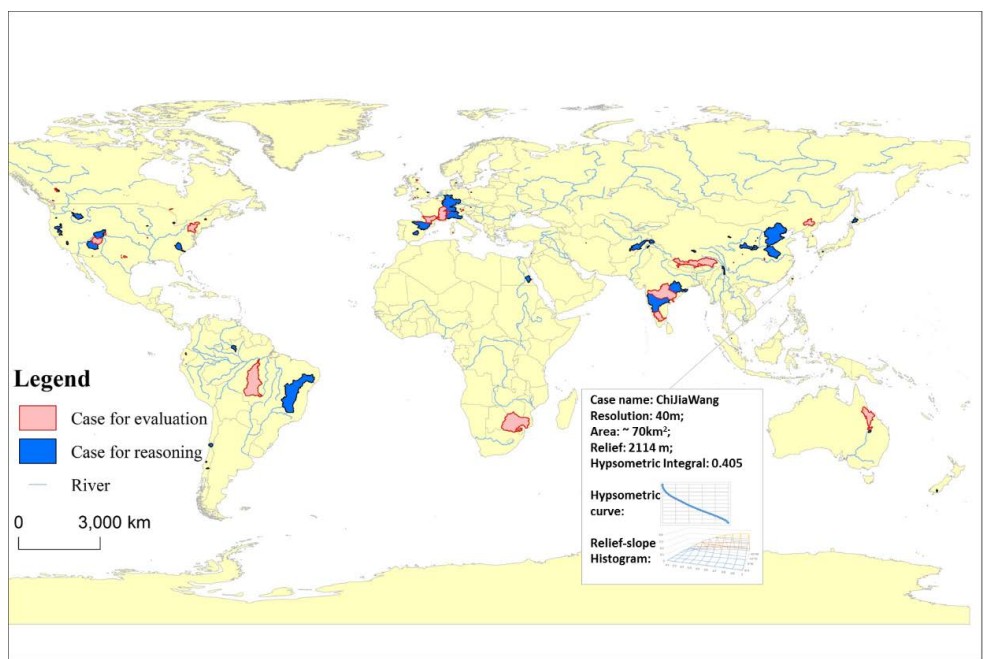

3   Figure 3. Spatial distribution of the cases used in this study (the box in the map shows an

4   example of a formalized case).





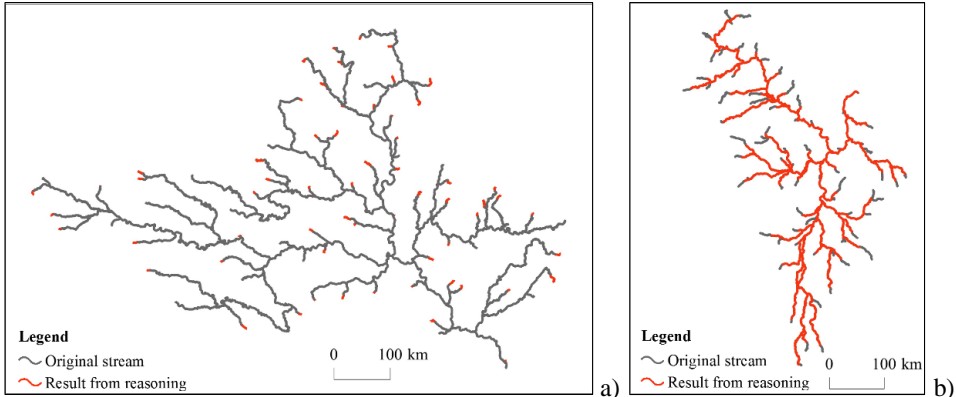

3  Figure 4. Comparison between the original drainage network of an individual evaluation case

4  and its extraction result using case-based reasoning: a) Godavari; and b) Burdekin.



## 1  Appendix. List of cases

| Case name | Source paper |
|-----------|--------------|
| LittlePiney | Botter G. Flow regime shifts in the Little Piney creek (US)[J]. Advances in Water Resources, 2014, 71: 44-54. |
| PoRiver | Lanzoni S, Luchi R, Pittaluga M B. Modeling the morphodynamic equilibrium of an intermediate reach of the Po River (Italy)[J]. Advances in Water Resources, 2015, 81: 95–102. |
| UpperMcKenzie | Di Lazzaro M, Zarlenga A, Volpi E. Hydrological effects of within-catchment heterogeneity of drainage density[J]. Advances in Water Resources, 2015, 76: 157-167. |
| Babaohe | Lei F, Huang C, Shen H, et al. Improving the estimation of hydrological states in the SWAT model via the ensemble Kalman smoother: Synthetic experiments for the Heihe River Basin in northwest China[J]. Advances in Water Resources, 2014, 67: 32-45. |
| OldMansCreek | Ayalew T B, Krajewski W F, Mantilla R, et al. Exploring the effects of hillslope-channel link dynamics and excess rainfall properties on the scaling structure of peak-discharge[J]. Advances in Water Resources, 2014, 64: 9-20. |
| UpstreamGarza | Balistrocchi M, Grossi G, Bacchi B. Deriving a practical analytical-probabilistic method to size flood routing reservoirs[J]. Advances in Water Resources, 2013, 62: 37-46. |
| Peacheater | Kim J, Warnock A, Ivanov V Y, et al. Coupled modeling of hydrologic and hydrodynamic processes including overland and channel flow[J]. Advances in Water Resources, 2012, 37: 104-126. |
| Cauvery | Konar M, Todd M J, Muneepeerakul R, et al. Hydrology as a driver of biodiversity: Controls on carrying capacity, niche formation, and dispersal[J]. Advances in Water Resources, 2013, 51: 317-325. |
| Krishna | |
| Krishna | |
| Godavari | |
| Klodawka | Jasiewicz J Ł, Metz M. A new GRASS GIS toolkit for Hortonian analysis of drainage networks[J]. Computers & Geosciences, 2011, 37(8): 1162-1173. |



| Chabagou | Li T, Wang G, Chen J. A modified binary tree codification of drainage networks to support complex hydrological models[J]. Computers & Geosciences, 2010, 36(11): 1427-1435. |
|---|---|
| SaoFrancisco<br>TapajosRiver | Saraiva A G S, Paz A R. Multi-step change of scale approach for deriving coarse-resolution flow directions[J]. Computers & Geosciences, 2014, 68: 53-63. |
| CooperRiver | Castronova A M, Goodall J L. A hierarchical network-based algorithm for multi-scale watershed delineation[J]. Computers & Geosciences, 2014, 72: 156-166. |
| MiddleColorado | Karimipour F, Ghandehari M, Ledoux H. Watershed delineation from the medial axis of river networks[J]. Computers & Geosciences, 2013, 59: 132-147. |
| FuRiver | Xu C, Xu X, Dai F, et al. Comparison of different models for susceptibility mapping of earthquake triggered landslides related with the 2008 Wenchuan earthquake in China[J]. Computers & Geosciences, 2012, 46: 317-329. |
| JuniataRiver<br>YoungWomansCreek | Yu X, Bhatt G, Duffy C, et al. Parameterization for distributed watershed modeling using national data and evolutionary algorithm[J]. Computers & Geosciences, 2013, 58: 80-90. |
| YaluTsangpo | Wang H, Fu X, Wang G. Multi-tree Coding Method (MCM) for drainage networks supporting high-efficient search[J]. Computers & Geosciences, 2013, 52: 300-306. |
| KaghanValley | Dehvari A, Heck R J. Removing non-ground points from automated photo-based DEM and evaluation of its accuracy with LiDAR DEM[J]. Computers & Geosciences, 2012, 43: 108-117. |
| CameronHighlands | Lim S L, Sagar B S D, Koo V C, et al. Morphological convexity measures for terrestrial basins derived from digital elevation models[J]. Computers & Geosciences, 2011, 37(9): 1285-1294. |
| W_Kharit | Milewski A, Sultan M, Yan E, et al. A remote sensing solution for estimating runoff and recharge in arid environments[J]. Journal of Hydrology, 2009, 373(1): 1-14. |
| ChiJiaWang<br>ErhWu | Lin W T, Chou W C, Lin C Y, et al. Automated suitable drainage network extraction from digital elevation models in Taiwan's upstream |




| | |
|---|---|
| | watersheds[J]. Hydrological Processes, 2006, 20(2): 289-306. |
| Demeni | Getirana A C V, Bonnet M P, Rotunno Filho O C, et al. Improving hydrological information acquisition from DEM processing in floodplains[J]. Hydrological Processes, 2009, 23(3): 502-514. |
| Batchawana | Creed I F, Hwang T, Lutz B, et al. Climate warming causes intensification of the hydrological cycle, resulting in changes to the vernal and autumnal windows in a northern temperate forest[J]. Hydrological Processes, 2015, 29: 3519–3534. |
| Hailogou | Xing B, Liu Z, Liu G, et al. Determination of runoff components using path analysis and isotopic measurements in a glacier‐covered alpine catchment (upper Hailuogou Valley) in southwest China[J]. Hydrological Processes, 2015, 29, 3065–3073. |
| Bellebeek | Loosvelt L, Pauwels V, Verhoest N E C. On the significance of crop‐type information for the simulation of catchment hydrology[J]. Hydrological Processes, 2015, 29(6): 915-926. |
| WeiRiver | Zuo D, Xu Z, Peng D, et al. Simulating spatiotemporal variability of blue and green water resources availability with uncertainty analysis[J]. Hydrological Processes, 2015, 29(8): 1942-1955. |
| HunzaRiver | Biber K, Khan S D, Shah M T. The source and fate of sediment and mercury in Hunza River basin, Northern Areas, Pakistan[J]. Hydrological Processes, 2015, 29(4): 579-587. |
| Kasilian | Saghafian B, Meghdadi A R, Sima S. Application of the WEPP model to determine sources of run‐off and sediment in a forested watershed[J]. Hydrological Processes, 2015, 29(4): 481-497. |
| Lonquen | Stewart R D, Abou Najm M R, Rupp D E, et al. Hillslope run‐off thresholds with shrink–swell clay soils[J]. Hydrological Processes, 2015, 29(4): 557-571. |
| MicaCreek1 MicaCreek2 | Du E, Link T E, Gravelle J A, et al. Validation and sensitivity test of the distributed hydrology soil‐vegetation model (DHSVM) in a forested mountain watershed[J]. Hydrological Processes, 2014, 28(26): 6196-6210. |
| NarayaniRiver | Neupane R P, Yao J, White J D. Estimating the effects of climate change on |



| | |
|---|---|
| | the intensification of monsoonal‐driven stream discharge in a Himalayan watershed[J]. Hydrological Processes, 2014, 28(26): 6236-6250. |
| WillowRiver Bowron | Zhang M, Wei X. Contrasted hydrological responses to forest harvesting in two large neighbouring watersheds in snow hydrology dominant environment: implications for forest management and future forest hydrology studies[J]. Hydrological Processes, 2014, 28(26): 6183-6195. |
| UpperDalya UpperTaninim | Peleg N, Shamir E, Georgakakos K P, et al. A framework for assessing hydrological regime sensitivity to climate change in a convective rainfall environment: a case study of two medium-sized eastern Mediterranean catchments, Israel[J]. Hydrology and Earth System Sciences, 2015, 19(1): 567-581. |
| SanFrancisco | Timbe E, Windhorst D, Crespo P, et al. Understanding uncertainties when inferring mean transit times of water trough tracer-based lumped-parameter models in Andean tropical montane cloud forest catchments[J]. Hydrology and Earth System Sciences, 2014, 18: 1503-1523. |
| HuaiRiver | Chen X, Hao Z, Devineni N, et al. Climate information based streamflow and rainfall forecasts for Huai River basin using hierarchical Bayesian modeling[J]. Hydrology and Earth System Sciences, 2014, 18(4): 1539-1548. |
| WarregoSC2 WarregoSC3 WarregoSC4 | Alvarez-Garreton C, Ryu D, Western A W, et al. Improving operational flood ensemble prediction by the assimilation of satellite soil moisture: comparison between lumped and semi-distributed schemes[J]. Hydrology and Earth System Sciences, 2015, 19(4): 1659-1676. |
| Ishikari | Duan W L, He B, Takara K, et al. Modeling suspended sediment sources and transport in the Ishikari River basin, Japan, using SPARROW[J]. Hydrology and Earth System Sciences, 2015, 19(3): 1293-1306. |
| Limari | Scott C A, Vicuña S, Blanco-Gutiérrez I, et al. Irrigation efficiency and water-policy implications for river basin resilience[J]. Hydrology and Earth System Sciences, 2014, 18(4): 1339-1348. |
| Limpopo | Trambauer P, Werner M, Winsemius H C, et al. Hydrological drought forecasting and skill assessment for the Limpopo River basin, southern Africa[J]. Hydrology and Earth System Sciences, 2015, 19(4): 1695-1711. |





| | |
|---|---|
| Crocodile<br><br>Komati | Saraiva Okello A M L, Masih I, Uhlenbrook S, et al. Drivers of spatial and temporal variability of streamflow in the Incomati River basin[J]. Hydrology and Earth System Sciences, 2015, 19(2): 657-673. |
| Haean | Shope C L, Maharjan G R, Tenhunen J, et al. Using the SWAT model to improve process descriptions and define hydrologic partitioning in South Korea[J]. Hydrology and Earth System Sciences, 2014, 18(2): 539-557. |
| Durance | Kuentz A, Mathevet T, Gailhard J, et al. Building long-term and high spatio-temporal resolution precipitation and air temperature reanalyses by mixing local observations and global atmospheric reanalyses: the ANATEM method[J]. Hydrology and Earth System Sciences, 2015, 19: 2717–2736. |
| Kabul | Wi S, Yang Y C E, Steinschneider S, et al. Calibration approaches for distributed hydrologic models in poorly gaged basins: implication for streamflow projections under climate change[J]. Hydrology and Earth System Sciences, 2015, 19(2): 857-876. |
| Garonne<br><br>Rhone | Habets F, Philippe E, Martin E, et al. Small farm dams: impact on river flows and sustainability in a context of climate change[J]. Hydrology and Earth System Sciences, 2014, 18(10): 4207–4222. |
| Ebro | Peñas F J, Barquín J, Snelder T H, et al. The influence of methodological procedures on hydrological classification performance[J]. Hydrology and Earth System Sciences, 2014, 18(9): 3393-3409. |
| Olifants | Dabrowski J M. Applying SWAT to predict orthophosphate loads and trophic status in four reservoirs in the upper Olifants catchment, South Africa[J]. Hydrology and Earth System Sciences, 2014, 18: 2629–2643. |
| WeiRiver | Zhan C S, Jiang S S, Sun F B, et al. Quantitative contribution of climate change and human activities to runoff changes in the Wei River basin, China[J]. Hydrology and Earth System Sciences, 2014, 18(8): 3069-3077. |
| Bellever<br><br>Brue<br><br>Bishop_Hull | Liu J, Han D. On selection of the optimal data time interval for real-time hydrological forecasting[J]. Hydrology and Earth System Sciences, 2013, 17(9): 3639-3659. |
| Pomahaka | McMillan H K, Hreinsson E Ö, Clark M P, et al. Operational hydrological data assimilation with the recursive ensemble Kalman filter[J]. Hydrology and Earth System Sciences, 2013, 17(1): 21-38. |





| ColoradoR_Cameron SanJuanR_Bluff DoloresR_Cisco | Rosenberg E A, Clark E A, Steinemann A C, et al. On the contribution of groundwater storage to interannual streamflow anomalies in the Colorado River basin[J]. Hydrology and Earth System Sciences, 2013, 17(4): 1475-1491. |
|---|---|
| RioSanFrancisco RioSanFrancisco | Windhorst D, Waltz T, Timbe E, et al. Impact of elevation and weather patterns on the isotopic composition of precipitation in a tropical montane rainforest[J]. Hydrology and Earth System Sciences, 2013, 17(1): 409-419. |
| Rhine | Vorogushyn S, Merz B. Flood trends along the Rhine: the role of river training[J]. Hydrology and Earth System Sciences, 2013, 17(10): 3871-3884. |
| Urola | Cowpertwait P, Ocio D, Collazos G, et al. Regionalised spatiotemporal rainfall and temperature models for flood studies in the Basque Country, Spain[J]. Hydrology and Earth System Sciences, 2013, 17: 479–494. |
| KrishnaRiver | Surinaidu L, Bacon C G D, Pavelic P. Agricultural groundwater management in the Upper Bhima Basin, India: current status and future scenarios[J]. Hydrology and Earth System Sciences, 2013, 17(2): 507-517. |
| ClearCreek | Zhang H L, Wang Y J, Wang Y Q, et al. The effect of watershed scale on HEC-HMS calibrated parameters: a case study in the Clear Creek watershed in Iowa, US[J]. Hydrology and Earth System Sciences, 2013, 17(7): 2735-2745. |
| Baba Toachi SanPabloLaMana | Arias-Hidalgo M, Bhattacharya B, Mynett A E, et al. Experiences in using the TMPA-3B42R satellite data to complement rain gauge measurements in the Ecuadorian coastal foothills[J]. Hydrology and Earth System Sciences, 2013, 17(7): 2905 |
| Monastir | Mascaro G, Piras M, Deidda R, et al. Distributed hydrologic modeling of a sparsely monitored basin in Sardinia, Italy, through hydrometeorological downscaling[J]. Hydrology and Earth System Sciences, 2013, 17(10): 4143-4158. |
| Gard | Braud I, Ayral P A, Bouvier C, et al. Multi-scale hydrometeorological observation and modelling for flash-flood understanding[J]. Hydrology and Earth System Sciences, 2014, 18(9): 3733-3761. |
| Zhanghe | Xie X, Meng S, Liang S, et al. Improving streamflow predictions at |





| | |
|---|---|
| | ungauged locations with real-time updating: application of an EnKF-based state-parameter estimation strategy[J]. Hydrology and Earth System Sciences, 2014, 18(10): 3923 |
| Davidson | Yang J, Castelli F, Chen Y. Multiobjective sensitivity analysis and optimization of distributed hydrologic model MOBIDIC[J]. Hydrology and Earth System Sciences, 2014, 18(10): 4101-4112. |
| Lienz | He Z H, Parajka J, Tian F Q, et al. Estimating degree-day factors from MODIS for snowmelt runoff modeling[J]. Hydrology and Earth System Sciences, 2014, 18(12): 4773-4789. |
| Cheakamus | Bourdin D R, Nipen T N, Stull R B. Reliable probabilistic forecasts from an ensemble reservoir inflow forecasting system[J]. Water Resources Research, 2014, 50(4): 3108-3130. |
| YbbsRiver | Ceola S, Bertuzzo E, Singer G, et al. Hydrologic controls on basin‐scale distribution of benthic invertebrates[J]. Water Resources Research, 2014, 50(4): 2903-2920. |
| Susquehanna | Giuliani M, Herman J D, Castelletti A, et al. Many‐objective reservoir policy identification and refinement to reduce policy inertia and myopia in water management[J]. Water Resources Research, 2014, 50(4): 3355-3377. |
| NorsmindeFjord | He X, Koch J, Sonnenborg T O, et al. Transition probability‐based stochastic geological modeling using airborne geophysical data and borehole data[J]. Water Resources Research, 2014, 50(4): 3147-3169. |
| SouthPark | Ball L B, Caine J S, Ge S. Controls on groundwater flow in a semiarid folded and faulted intermountain basin[J]. Water Resources Research, 2014, 50(8): 6788-6809. |
| KernRiver | Girotto M, Cortés G, Margulis S A, et al. Examining spatial and temporal variability in snow water equivalent using a 27 year reanalysis: Kern River watershed, Sierra Nevada[J]. Water Resources Research, 2014, 50(8): 6713-6734 |
| UpperRhone | Bordoy R, Burlando P. Stochastic downscaling of climate model precipitation outputs in orographically complex regions: 2. Downscaling methodology[J]. Water Resources Research, 2014, 50(1): 562-579. |
| Pettit | Mallard J, McGlynn B, Covino T. Lateral inflows, stream‐groundwater |





| Stanley | exchange, and network geometry influence stream water composition[J]. Water Resources Research, 2014, 50(6): 4603-4623. |
|---|---|
| Alturas | |
| Burdekin | Bainbridge Z T, Lewis S E, Smithers S G, et al. Fine‐suspended sediment and water budgets for a large, seasonally dry tropical catchment: Burdekin River catchment, Queensland, Australia[J]. Water Resources Research, 2014, 50(11): 9067-9087. |
| Blackwater | Cooper R J, Krueger T, Hiscock K M, et al. Sensitivity of fluvial sediment source apportionment to mixing model assumptions: A Bayesian model comparison[J]. Water Resources Research, 2014, 50(11): 9031-9047. |
| OitaRiver | Higashino M, Stefan H G. Modeling the effect of rainfall intensity on soil‐water nutrient exchange in flooded rice paddies and implications for nitrate fertilizer runoff to the Oita River in Japan[J]. Water Resources Research, 2014, 50(11): 8611-8624. |
| Zwalm | Guingla P, Douglas A, Keyser R, et al. Improving particle filters in rainfall‐runoff models: Application of the resample‐move step and the ensemble Gaussian particle filter[J]. Water Resources Research, 2013, 49(7): 4005-4021. |
| XianNanGou | Ichoku C, Karnieli A, Verchovsky I. Application of fractal techniques to the comparative evaluation of two methods of extracting channel networks from digital elevation models[J]. Water Resources Research, 1996, 32(2): 389-399. |
| Hodder | Bulygina N, Ballard C, McIntyre N, et al. Integrating different types of information into hydrological model parameter estimation: Application to ungauged catchments and land use scenario analysis[J]. Water Resources Research, 2012, 48(6), W06519. |
| NorthEsk | Capell R, Tetzlaff D, Soulsby C. Can time domain and source area tracers reduce uncertainty in rainfall‐runoff models in larger heterogeneous catchments?[J]. Water Resources Research, 2012, 48(9), W09544. |
| SouthForkNew | Gu C, Anderson W, Maggi F. Riparian biogeochemical hot moments induced by stream fluctuations[J]. Water Resources Research, 2012, 48(9), W09546. |





| LiWuRiver | Huang Jr C, Yu C K, Lee J Y, et al. Linking typhoon tracks and spatial rainfall patterns for improving flood lead time predictions over a mesoscale mountainous watershed[J]. Water Resources Research, 2012, 48(9), W09540. |
|---|---|
| AlzetteEttel | Krier R, Matgen P, Goergen K, et al. Inferring catchment precipitation by doing hydrology backward: A test in 24 small and mesoscale catchments in Luxembourg[J]. Water Resources Research, 2012, 48(10), W10525. |
| MessPontpierre | |
| Colpach | |
| RoudbachPlaten | |
| Burdekin | Kuhnert P M, Henderson B L, Lewis S E, et al. Quantifying total suspended sediment export from the Burdekin River catchment using the loads regression estimator tool[J]. Water Resources Research, 2012, 48(4), W04533. |
| Cajon | Mendoza P A, McPhee J, Vargas X. Uncertainty in flood forecasting: A distributed modeling approach in a sparse data catchment[J]. Water Resources Research, 2012, 48(9), W09532. |
| Tenderfoot | Payn R A, Gooseff M N, McGlynn B L, et al. Exploring changes in the spatial distribution of stream baseflow generation during a seasonal recession[J]. Water Resources Research, 2012, 48(4), W04519. |
| Wattenbach | Rogger M, Pirkl H, Viglione A, et al. Step changes in the flood frequency curve: Process controls[J]. Water Resources Research, 2012, 48(5), W05544. |
| Weerbach | |
| UpperRhone | Leite Ribeiro M, Blanckaert K, Roy A G, et al. Hydromorphological implications of local tributary widening for river rehabilitation[J]. Water Resources Research, 2012, 48(10), W10528. |
| WhiteRiver | Steinschneider S, Polebitski A, Brown C, et al. Toward a statistical framework to quantify the uncertainties of hydrologic response under climate change[J]. Water Resources Research, 2012, 48(11), W11525. |
| AmericanRiver | Woldemichael A T, Hossain F, Pielke R, et al. Understanding the impact of dam‐triggered land use/land cover change on the modification of extreme precipitation[J]. Water Resources Research, 2012, 48(9), W09547. |
| MahanadiRiver | Kannan S, Ghosh S. A nonparametric kernel regression model for |



| | |
|---|---|
| | downscaling multisite daily precipitation in the Mahanadi basin[J]. Water Resources Research, 2013, 49(3): 1360-1385. |
| Nujiang | Kibler K M, Tullos D D. Cumulative biophysical impact of small and large hydropower development in Nu River, China[J]. Water Resources Research, 2013, 49(6): 3104-3118. |
| LuckyHills | Sivandran G, Bras R L. Dynamic root distributions in ecohydrological modeling: A case study at Walnut Gulch Experimental Watershed[J]. Water Resources Research, 2013, 49(6): 3292-3305. |
| Sacramento Feather | Ficklin D L, Stewart I T, Maurer E P. Effects of climate change on stream temperature, dissolved oxygen, and sediment concentration in the Sierra Nevada in California[J]. Water Resources Research, 2013, 49(5): 2765-2782. |
| ClintonRiver | Shen C, Niu J, Phanikumar M S. Evaluating controls on coupled hydrologic and vegetation dynamics in a humid continental climate watershed using a subsurface‐land surface processes model[J]. Water Resources Research, 2013, 49(5): 2552-2572. |
| HJA | Garcia E S, Tague C L, Choate J S. Influence of spatial temperature estimation method in ecohydrologic modeling in the Western Oregon Cascades[J]. Water Resources Research, 2013, 49(3): 1611-1624. |
| UpperGuadiana | Loon A F, Lanen H A J. Making the distinction between water scarcity and drought using an observation‐modeling framework[J]. Water Resources Research, 2013, 49(3): 1483-1502. |
| HaiRiver | Jia Y, Ding X, Wang H, et al. Attribution of water resources evolution in the highly water‐stressed Hai River Basin of China[J]. Water Resources Research, 2012, 48(2), W02513. |
| Cordevole | Rigon E, Comiti F, Lenzi M A. Large wood storage in streams of the Eastern Italian Alps and the relevance of hillslope processes[J]. Water Resources Research, 2012, 48(1), W01518. |
| SalmonRiver | Yearsley J. A grid‐based approach for simulating stream temperature[J]. Water Resources Research, 2012, 48(3), W03506. |
| CedoCaka | Zhang G, Xie H, Yao T, et al. Snow cover dynamics of four lake basins |



| YamzhogYumCo | over Tibetan Plateau using time series MODIS data (2001–2010)[J]. Water Resources Research, 2012, 48(10), W10529. |
|---|---|

