# Peer review of "Case-based knowledge formalization and reasoning method for digital terrain analysis — Application to extracting drainage networks"

_Hydrology and Earth System Sciences, 2015_

## Short Comment (SC1) · 20 Mar 2016

The paper proposed a formalization and reasoning method for knowledge sharing during model reusing process. Although a little bit simply, the idea is very interesting, and may be extended to similar research on geographic modeling and model integration. Here, some comments for your considering. 1. In your paper, as determining the CA is a simple function related to DTA and DTA is also just a part of modeling method, how to deal with the complexity problem when conducting comprehensive research and analysis, i.e., how to formalized the complex knowledge about some complex problems, the

semantic problem, the structure to represent the knowledge, ect. Do you have some preliminary ideas? I think this is the key step to promote your idea into application in a broader field. 2. In page 2, you mentioned that "However, current DTA-assisted tools...provide very limited support during the DTA application modeling process". The conclusion is somewhat arbitrary, you may need to provide more arguments here. 3. page 3, line 6, "largely inaccurate" ïij§ 4. page 4, line 19, "is not necessary to participate", why? please explain it clearly. 5. Page 7, part 4.1, I think maybe you need to provide a table here to explain your quantitative attributes, not just some sentences. 6. Page 13, line 2, how to realize your "automatic program" to derive other attributes? Do you mean that these attributes have been processed into a dataset? Otherwise, I think it is hard to automatic match these attributes in their text manual. 7. Page 14, line23, 0.4->0.43. 8. Page 20, table 2, do you consider some other parameters? For example, I think the characteristics of study area are somewhat simple. 9. Page 23, the overall similarity, can it be calculated using weighting? 10. Figure 4, part b. is it right?

---

## Referee Comment (RC1) · Anonymous Referee #1 · 15 May 2016

The paper presents an interesting and potentially valuable approach to suggest parameters for stream extraction tools based on the past use cases. Although the manuscript is generally well structured there are issues with the use of terminology and description of the methods and use cases.

Specifically:

p. 1 l. 19 DTA-assisted tools (e.g., ArcGIS, GRASS, SAGA, White Box, TauDEM) ArcGIS and GRASS are large, general purpose GIS packages which include DTA tools, - reference to specific modules is needed here.

l. 25 I find the following sentence confusing algorithm knowledge, which is the meta-data of a DTA algorithm - what do authors mean by this?

p.2 l. 1 again ModelBuilder is not DTA-assisted tool - not clear what is meant here

p.3, l. 6 this assumes that there is no validation data available - isn't the best way to find the optimal parameters running the tools with a set of parameters and find the best fit with the field data (or remotely sensed data if they provide sufficient information)? What if the case studies are inaccurate? Can this be taken into account?

p. 7 l. 15 What is meant by aspect here?

l. 17 how do you compute relief - you refer to it as steep or gently sloping - isn't that equivalent to slope? Relief in gemorphometry is a very specific metrics - specify here what you are using or use different term

l. 24 seven grades? did you meant seven classes or categories? It appears that you mix relief and slope - perhaps use equations to precisely define what you mean

l. l. 26-27 10 level x 7 grade - did you mean 10 elevation classes x 7 slope classes?

l. 30 relieves the DEM resolution effect ? what do you mean by relieves?

p. 8 l. 20 comment - environmental conditions, especially the groundwater level could be more important than the topo parameters, so the case studies used should be evaluated for this and those where parameters other than the proposed ones play determining role should be excluded

p. 9 l. 17 - Doesn't the need to empirically adjust the shape of the bell curve beat the purpose of the proposed method?

eq. 1 ln(0.5) is a constant - why ln and not the constant value directly?

p. 10 l. 4 and 5 magnitude of cell size - did you mean absolute value? magnitude does not make sense here. If it is indeed absolute value (as indicated in Table 2), this

treats cell size larger the same as cell size smaller - there is a fundamental difference between downscaling and upscaling or going to higher level of detail versus lower level of detail in terms of stream extraction - how do you account for this issue?

p. 10 l. 15 - what is meant by area - total area of the study site? magnitude here probably should be again the absolute value

p. 10 l. 22 it is not clear what is meant by relief here - providing an equation or more precise definition is necessary, is it the difference between the minimum and maximum elevation in the study area? If yes, please check how the term relief is used in literature and what should you use here.

p. 12 l. 1 the presented workflow applies to only the older algorithms and is highly simplified - this needs to be mentioned. For example, filling of pits (many are often real) and flat areas is not necessary if least cost path algorithm is used - see e.g. Metz et al. 2011, doi:10.5194/hess-15-667-2011r the second step also is not quite accurate - spatial distribution of catchment area sounds confusing - perhaps you meant flow accumulation or contributing areas for each grid cell?

l. 15 it is apparent that the proposed experiment applies only to ArcGIS-based workflow which is highly limited and somewhat obsolete, but it can still be used as a case study, given the large number of users who would use this tool. Were all the articles used as case base using the same algorithm?

p. 12 l. 29, 30 - what is meant by extracting here? perhaps identifying?

did all articles use SRTM or ASTER?

It is not clear why river density for evaluations - how is it computed? i Why not the total length of the river network? How many validation cases lead to shorter streams and how many were longer (see Fig. 4).

Overall this is a promising approach, proposed in a highly simplified form in the manuscript. Careful revision of terminology and clarification of the workflow and algorithm issues is needed to avoid confusion and make the paper scientifically sound.

---

## Referee Comment (RC2) · Anonymous Referee #2 · 29 May 2016

Formalization of application context knowledge in digital terrain analysis I feel as relevant scientific problem also within the scope of HESS. The paper presents a novel concept of such formalization supported by experiment based on elaboration of results of 124 relevant papers. The overall presentation is well structured and clear. Amount and quality of supplementary material is appropriate. However I have a doubt about method used. Not that I contest the learning from previous studies, but procedure suggested in the paper contain too many debatable aspects.

Principal problems i) Authors suggest to replace a deep functional analysis of application context by the method based on learning from various previous solutions regardless of their detailed knowledge. OK, deeper functional analysis can be too difficult and selection of only some elements of application context can be a solution. However selection of used attributes and similarity functions was reasoned only poorly and in no way verified. ii) Presumption that articles published in good journals are supposed to provide good solutions for their specific study areas based on experts' experience and knowledge of the target task can be justified in general, but it is probably too optimistic in some cases even considering that determination of drainage network is probably only marginal problem for a part of articles. So no every solution published in good journal have to be well. And therefore a method based on selection the only one 'exemplary' published solution I feel as problematic. iii) While the suggested computation of similarity of individual attributes between the new application and published one can be acceptable, the synthesis (computation of 'overall similarity') is more problematic. No (equal) weighting of used attributes is a basic problem. It is very improbable that similarity in name of target task, cell size, area, relief, slope distribution and hypsometric integral will have the same effect on determination of proper catchment area threshold for extracting drainage networks. iv) Evaluation of experimental results is very problematic. Authors write (23-25, p.13): "Four levels of E were established empirically to reflect the reasonableness level: reasonable (E$\in$[0,0.1]), acceptable (E$\in$(0.1,0.25]), questionable (E$\in$(0.25,0.5]), and unreasonable (E$\in$(0.5,+$\infty$))." It is non committal for me and if authors do not specify this 'empirical establishment' I feel it as fully subjective division. Why the difference in drainage density is unreasonable only exceed 50 %?! It smell by purpose made establishment of intervals to show "that the proposed method performs satisfactorily" (9, p.14).

Some another problems / ambiguities - The title of the paper is too complex and not quite clear. A simplification is suitable (e.g. Case-based formalization and reasoning method for digital terrain analysis â˘Tà determining the catchment area threshold for extracting drainage networks). - Because equal weight of all attributes the binary attribute 'the name of the target task' exclude (in final comparison) all cases with another name of the target task. What is the reason of such hard limit? How was determined the attribute for particular cases? Names of types and their occurrence should be added for better understanding. - Attribute relief - is it one number for the whole area (then it very dependents on area size) or average value computed by what way? (moving window - the size and shape?) - Slope is scale dependent variable so distribution of slopes depend on grid size. Using of cumulative frequency distribution solve this problem only partially. - Similarity functions seem to be determined subjectively. Why difference in magnitude of cell size (and area) can better reflect the level of similarity between DTA applications than the numerical difference in cell size? Why is used natural log in one case and common in another? etc.

Final evaluation and suggestion In regard to aforementioned problems I cannot recommend the paper in the present form, (presented experiment is not enough documented to support the interpretations and conclusions). However, majority of problems could be eliminated by selection of more appropriate method of synthesis. I think, multidimensional regression is a way. This method provide for elimination of inappropriate possible influence of particular problematic published case studies (ii), reveal various weights (suitability) of used attributed and similarity functions (mainly if hierarchical partitioning will be used) (iii) and last but not least alternative results (using various attributes and methods of similarity computation) can be compared to find the most appropriate regression equation. Suitability of selected attributes and methods can be documented by this way (i) and it can partly also substitute problematic way of evaluation in this paper (iv).

Please also note the supplement to this comment:
http://www.hydrol-earth-syst-sci-discuss.net/hess-2015-539/hess-2015-539-RC2-supplement.pdf

**Supplement:**

[revised manuscript text omitted]

---

## Author Comment (AC1) · 19 Jun 2016

The authors thank Dr. M. Chen for the constructive comments which are helpful for improving the final version of this paper. We answer these comments as below.

Comment 1: In your paper, as determining the CA is a simple function related to DTA and DTA is also just a part of modeling method, how to deal with the complexity problem when conducting comprehensive research and analysis, i.e., how to formalized the complex knowledge about some complex problems, the semantic problem, the structure to represent the knowledge, ect. Do you have some preliminary ideas? I think this

[Figure]

is the key step to promote your idea into application in a broader field.

Response: This study explores how to formalize application-context knowledge in DTA and apply it to DTA modeling, when other two types of DTA knowledge (i.e., task knowledge and algorithm knowledge) have been formalized by means of rule or semantic networks (Russell and Norvig, 2009) and hence can be used in existing DTA-assisted tools. Combining the propose method with existing methods for using other two types of DTA knowledge, automated DTA modeling could be implemented to make DTA easy to use for users (especially non-expert users) and ensure that the result model is reasonable comparatively. For other geographic modeling domains, normally the modeling knowledge could also be classified into these three types, i.e., task knowledge, algorithm knowledge, and application-context knowledge. The task and algorithm knowledge in some domains (e.g., watershed modeling) which are more complex than those in DTA have been explored for formalization and inference methods and corresponding tools, such as Gregersen et al. (2007) and Škerjanec et al. (2014) in automated watershed modeling domain. For those geographic modeling domains in which the application context knowledge is also largely non-systematic and tacit knowledge, the case-based idea proposed in this manuscript could also be available to combining with the existing automated modeling methods of using the task and algorithm knowledge in these domains. We will revise the manuscript to include the discussion on this issue.

Comment 2: In page 2, you mentioned that "However, current DTA-assisted tools. . .provide very limited support during the DTA application modeling process". The conclusion is somewhat arbitrary, you may need to provide more arguments here.

Response: Currently, there is no well-established formalization method for application-context knowledge. Existing DTA-assisted tools, which have used the task knowledge and algorithm knowledge, consequently cannot use this type of knowledge to provide more effective support to DTA application modeling process. This situation exists mainly because this type of DTA knowledge is largely non-systematic and tacit knowledge, and often exists only in documents for specific case studies (DTA application instances) or even just in the experience of domain experts. We will revised the manuscript to state this point.

Comment 3: page 3, line 6, "largely inaccurate"

Response: The application-context knowledge of DTA is is largely non-systematic and tacit knowledge. We will revise the manuscript accordingly.

Comment 4: page 4, line 19, "is not necessary to participate", why? please explain it clearly.

Response: Only the problem part of each case is used to calculate the similarity between the case and the new application problem. The solution of the case with the highest similarity is retrieved as the solution for the new DTA application problem. Thus the solution part of a case does not participate in the reasoning procedure. We will revise the manuscript to state this point.

Comment 5: Page 7, part 4.1, I think maybe you need to provide a table here to explain your quantitative attributes, not just some sentences.

Response: Table 2 lists the attributes used to formalize a case problem and the corresponding similarity functions used in the proposed method.

Comment 6: Page 13, line 2, how to realize your "automatic program" to derive other attributes? Do you mean that these attributes have been processed into a dataset? Otherwise, I think it is hard to automatic match these attributes in their text manual.

Response: In this study, we manually selected the peer-reviewed papers related to the drainage network extraction applications which were published in mainstream journals of related domains. After the study area of each case was set, an automatic program was applied to SRTM DEM or ASTER GDEM of the study area to derive attributes (such as area, total relief, elevation-slope cumulative frequency distribution, and hypsometric curve) for each case. The results were recorded in the case base. We will make it clear in the revised manuscript.

**HESSD**

Comment 7: Page 14, line23, 0.4->0.43.

Response: We will revise the manuscript to correct it.

Comment 8: Page 20, table 2, do you consider some other parameters? For example, I think the characteristics of study area are somewhat simple.

Response: The method proposed in current study focuses on DTA domain and considers the area and the terrain condition through a few simple attributes for describing the study area characteristics of a DTA application case. Preliminary evaluation results show the reasonableness of the proposed method. The design of the attributes used to describe the problem part of a case could be improved to describe the domain-specific application-context information in an all-round and efficient manner, which needs additional research. We will revise the manuscript to discuss this issue.

Comment 9: Page 23, the overall similarity, can it be calculated using weighting?

Response: In current method proposed, the overall similarity between a case and a new application problem is determined by applying a minimum operator to synthesizing the similarity values on every attribute in a cautious manner. In the geographical modeling domain, a minimum operator based on the limiting factor principle is often used to synthesize similarity values on multiple attributes (Zhu and Band, 1994). The overall similarity result by a minimum operator is lower (i.e., higher uncertainty of reasoning result) than those from other synthesis means such as weighted-average. Based on the experiment shown in the original manuscript, we also test the effect of calculating the overall similarity by a simple average operator (a representative of weighted-average) instead of the minimum operator. The evaluation results show that the overall similarity for every case increased and the lowest overall similarity among results for 50 evaluation cases increased from 0.47 to 0.68 when the minimum operator was replaced by the simple average operator. Among 50 evaluation cases, the solutions for 13 evaluation cases from the proposed method changed because the cases with the highest similarity resulted by the simple average operator were different from those resulted by

the minimum operator. Due to the synthesis by the simple average operator instead of the minimum operator, the relative deviation of river density (E) increased for 10 of these 13 evaluation cases with different solutions, when E slightly decreased for other 3 evaluation cases. The increase of E even reached 20~80 times for some cases with the overall similarity values larger than 0.8. Because the overall similarity values were larger than 0.8 for most of evaluation cases, there is no a reasonable relationship between the overall similarity value and the E. This shows that the proposed method performed poorly when the simple average operator was used instead of the minimum operator. Note that the simple average is the common representative of weighted-average, and currently it is difficult to choose a more complex weighted-average for synthesizing similarity values on multiple attributes. Therefore the synthesis by a minimum operator is proposed for current method in this study. Additional research is needed to evaluate the similarity calculation method through further test with more types of DTA applications. We will revise the manuscript to include above discussion.

Comment 10: Figure 4, part b. is it right?

Response: Fig. 4b is correct. For this case, the CA threshold resulted from the proposed method is larger than it recorded in the evalution case, which means that the drainage network extracted by using the the CA threshold result is shorter than the original drainage network of this case. The situation shown in Fig. 4a is contrary.

References

Gregersen, J. B., Gijsbers, P. J. A., and Westen, S. J. P.: OpenMI: Open modelling interface, J. Hydroinfo., 9(3), 175-191, 2007.

Škerjanec, M., Atanasova, N., Cerepnalkoski, D, Dzeroski, S., and Kompare, B.: Development of a knowledge library for automated watershed modeling, Environ. Modell. Softw., 54, 60-72, 2014.

Russell, S. and Norvig, P.: Artificial Intelligence: a Modern Approach (3rd Edition),

Prentice Hall, 2009.

Zhu, A-X. and Band, L.: A knowledge-based approach to data integration for soil mapping, Can. J. Remote Sens., 20, 408-418, 1994.
* * *

---

## Author Comment (AC2) · 19 Jun 2016

The authors thank the anonymous referee for the constructive comments which are helpful for improving the final version of this manuscript. We answer these comments as below.

Comment 1: p. 1 l. 19 DTA-assisted tools (e.g., ArcGIS, GRASS, SAGA, White Box, TauDEM) ArcGIS and GRASS are large, general purpose GIS packages which include DTA tools, - reference to specific modules is needed here.

Response: DTA-assisted tools include general purpose GIS packages with DTA func-

tionality (e.g., "Spatial Analyst" toolbar in ArcGIS, r.* modules in GRASS, "Terrain Analysis" menu in SAGA, etc.) and domain-specific software (e.g., Whitebox, TauDEM, etc.) (Hengl and Reuter, 2009). We will revise the manuscript to clarify this point.

Comment 2: l. 25 I find the following sentence confusing algorithm knowledge, which is the metadata of a DTA algorithm - what do authors mean by this?

Response: The algorithm knowledge is the metadata of a DTA algorithm (including its parameters), such as the data type of input/output file, the number of parameters, and the valid range for each parameter. We will revise the manuscript to clarify this point.

Comment 3: p.2 l. 1 again ModelBuilder is not DTA-assisted tool - not clear what is meant here Response: ModelBuilder module in ArcGIS uses task knowledge and algorithm knowledge to aid connecting a set of DTA algorithms to be an executable DTA workflow in a interactive visual way. We will revise the manuscript to make it clear.

Comment 4: p.3, l. 6 this assumes that there is no validation data available - isn't the best way to find the optimal parameters running the tools with a set of parameters and find the best fit with the field data (or remotely sensed data if they provide sufficient information)? What if the case studies are inaccurate? Can this be taken into account?

Response: We agree with the reviewer that the best way to determine the optimal parameter-settings should be the evaluation based on the field data. However, at the beginning of the modeling, field data might be not easy to be obtained and the evaluation process is not easy to operate for those non-expert users. The method proposed in this study might automate the DTA modeling process, which makes it easy for users (especially non-expert users), and meanwhile the result model could be reasonable comparatively. We will revise the manuscript to discuss this point. The algorithm and parameter-settings presented in those journal papers might not be optimal, thus the corresponding cases might be inaccurate. In this study, we manually selected the peer-reviewed papers related to the drainage network extraction applications which were published in mainstream journals of related domains. By this means the cases

used could be kept as accurate (or reliable) as possible. Additional research is needed to enhance the proposed method by taking the reliability of the case into account. We will revise the manuscript to discuss this point.

Comment 5: p. 7 l. 15 What is meant by aspect here?

Response: Here the "aspect" means the kind of attributes designed to describe the terrain condition. We will revise the manuscript to use the term unambiguously.

Comment 6: l. 17 how do you compute relief - you refer to it as steep or gently sloping - isn't that equivalent to slope? Relief in geomorphometry is a very specific metrics - specify here what you are using or use different term

Response: Here it means the total relief of the study area, which is the maximum minus minimum elevation within the study area. We will revise the manuscript to use the term properly.

Comment 7: l. 24 seven grades? did you meant seven classes or categories? It appears that you mix relief and slope - perhaps use equations to precisely define what you mean

Response: Yes, the slope gradient value was divided into seven classes. We will revise the manuscript to make it clear and also precisely define the calculation of the total relief used in this study.

Comment 8: l. 26-27 10 level x 7 grade - did you mean 10 elevation classes x 7 slope classes?

Response: Yes, we will revise the manuscript to make it clear and use the term "elevation-slope cumulative frequency distribution" instead of the "relief-slope cumulative frequency distribution" used in the original manuscript.

Comment 9: l. 30 relieves the DEM resolution effect ? what do you mean by relieves?

Response: DEM resolution has a strong influence on calculating the slope gradient and

its frequency distribution (Chang and Tsai, 1991; Grohmann, 2015), while the DEM resolution has a comparatively weak influence on the cumulative frequency distribution of slope gradient. To relieve this DEM resolution effect and ensure the comparability of slope distributions from two cases with different DEM resolutions, we use the slope cumulative frequency in this study instead of the slope frequency distribution to describe the slope distribution. We will revise the manuscript to clarify this point.

Comment 10: p. 8 l. 20 comment - environmental conditions, especially the groundwater level could be more important than the topo parameters, so the case studies used should be evaluated for this and those where parameters other than the proposed ones play determining role should be excluded

Response: We agree with the reviewer that the groundwater level also plays important role on drainage network formation. However, the information of groundwater level is often difficult to be collected. Normal way of drainage network extraction by DTA is mainly based on topographic information. The method proposed in current study focuses on DTA domain and considers the area and the terrain condition for describing the study area characteristics of a DTA application case. Preliminary evaluation results show the reasonableness of the proposed method. The design of the attributes used to describe the problem part of a case could be improved to describe the domain-specific application-context information in an all-round and efficient manner, which needs additional research. We will revise the manuscript to discuss this issue.

Comment 11: p. 9 l. 17 - Doesn't the need to empirically adjust the shape of the bell curve beat the purpose of the proposed method?

Response: Currently we empirically set the shape-adjusting parameter (w) with fixed values for two attributes with bell-shaped similarity function. Preliminary evaluation results show that the proposed method with these settings performs well. The way of setting the shape-adjusting parameter will be explored as a part of future research. For example, if case base is with a large size, a machine learning algorithm might be

available for calibrating the shape-adjusting parameter automatically. We will revise the manuscript to include the discussion on this issue.

Comment 12: eq. 1 ln(0.5) is a constant - why ln and not the constant value directly?

Response: We accept this advice and will revise Eq. (1) accordingly.

Comment 13: p. 10 l. 4 and 5 magnitude of cell size - did you mean absolute value? Magnitude does not make sense here. If it is indeed absolute value (as indicated in Table 2), this treats cell size larger the same as cell size smaller - there is a fundamental difference between downscaling and upscaling or going to higher level of detail versus lower level of detail in terms of stream extraction - how do you account for this issue?

Response: In this study, we try to keep the similarity function on each attribute as a simpler form before more detailed research could be conducted to improve it. Current design of the similarity function for cell size is mainly based on two reasons. First, the numerical difference in cell size does not work. Taking an application with 10-m resolution as example, another application with a coarser resolution of 25 m is comparable to it from a cell size perspective, while on the other hand the resolution cannot be less than or equal to 0 m. Secondly, a bell-shaped similarity function for a logarithmic transformation of cell size could balance the decrease of similarity value for those situations with a coarser resolution or a finer resolution. Note that the similarity value on cell size will rapidly decrease to be about 0.58 when the resolution is coarsened to be double the resolution of a case or is refined to be a half of the case's resolution. The lower similarity value will deny the corresponding case to be a credible solution provider for the new application problem. This means that the current method proposed does not suggest a large-step downscaling and upscaling application of existing cases. We will revise the manuscript to state this point.

Comment 15: p. 10 l. 15 - what is meant by area - total area of the study site? magnitude here probably should be again the absolute value

Response: Yes, the area attribute is the total area of the study site. In this study we design a bell-shaped similarity function for a logarithmic transformation of area based on the idea similar to the design for the cell size attribute. Please also see our response above to the 14th item of comments from anonymous referee #1. We will revise the manuscript to make this point clear.

Comment 16: p. 10 l. 22 it is not clear what is meant by relief here - providing an equation or more precise definition is necessary, is it the difference between the minimum and maximum elevation in the study area? If yes, please check how the term relief is used in literature and what should you use here.

Response: We will revise the manuscript to use the term "total relief" and also precisely define the calculation of the total relief, i.e., the maximum minus minimum elevation within the study area.

Comment 17: p. 12 l. 1 the presented workflow applies to only the older algorithms and is highly simplified - this needs to be mentioned. For example, filling of pits (many are often real) and flat areas is not necessary if least cost path algorithm is used - see e.g. Metz et al. 2011, doi:10.5194/hess-15-667-2011r the second step also is not quite accurate - spatial distribution of catchment area sounds confusing - perhaps you meant flow accumulation or contributing areas for each grid cell?

Response: We accept this advice and will revise the manuscript accordingly. A new reference (Metz, M., Mitasova, H., and Harmon, R. S.: Efficient extraction of drainage networks from massive, radar-based elevation models with least cost path search, Hydrol. Earth Syst. Sci., 15, 667-678, 2011) will be cited in the revised manuscript.

Comment 18: l. 15 it is apparent that the proposed experiment applies only to ArcGIS-based workflow which is highly limited and somewhat obsolete, but it can still be used as a case study, given the large number of users who would use this tool. Were all the articles used as case base using the same algorithm?

Response: In most of articles used for case preparation a single flow direction algorithm (such as D8 algorithm) was adopted, when a few articles did not state clearly the flow direction algorithm used. Note that the experiment in this study was designed to focus on the determination of CA threshold for drainage network extraction, not the flow direction algorithm used. We will revise the manuscript to state this point.

Comment 19: p. 12 l. 29, 30 - what is meant by extracting here? perhaps identifying?

Response: Yes, we will revise the manuscript accordingly.

Comment 20: did all articles use SRTM or ASTER?

Response: Some articles used for case preparation in this study used DEM with a finer resolution than that of SRTM or ASTER DEM. However, those DEM are often not easy to collect by us. Therefore, we used these open DEM data to derive the case attributes such as area, total relief, elevation-slope cumulative frequency distribution, and hypsometric curve. And this process also makes each of these attributes comparable between a case and a new application problem. We will state this point in the revised manuscript.

Comment 21: It is not clear why river density for evaluations - how is it computed? Why not the total length of the river network? How many validation cases lead to shorter streams and how many were longer (see Fig. 4).

Response: The river density was calculated by the total length of the extracted drainage network divided by the area of the study site. In current manuscript, the relative deviation of river density was used as an index for quantitative evaluation of the proposed method. Based on Eq. (2) in the manuscript, which defines this index, the index value will be same if the total length of river network is used instead of the river density. Compared with the length of drainage network, the river density can also be used to make comparison between the results for different application problems, although this comparison has not been made for discussion in current manuscript. The

counts of validation cases which got shorter and longer drainage networks from the proposed method are 16 and 28, respectively. We will revise the manuscript to provide this information.

References

Chang, K. and Tsai, B.: The effect of DEM resolution on slope and aspect mapping, Cartogr. Geogr. Inf. Syst., 18, 69-77, 1991.

Grohmann, C. H.: Effects of spatial resolution on slope and aspect derivation for regional-scale analysis, Comput. Geosci., 77, 111-117, 2015.

Hengl, T. and Reuter, H. I.: Geomorphometry: Concepts, Software, Applications, Elsevier, Amsterdam, 2009.

---

## Author Comment (AC3) · 20 Jun 2016

The authors thank the anonymous referee for the constructive comments which are helpful for improving the final version of this manuscript. We answer these comments as below.

Comment 1: Authors suggest to replace a deep functional analysis of application context by the method based on learning from various previous solutions regardless of their detailed knowledge. OK, deeper functional analysis can be too difficult and selection of only some elements of application context can be a solution. However selection

of used attributes and similarity functions was reasoned only poorly and in no way verified.

Response: In this study we explored how to formalize application-context knowledge in DTA and apply it to DTA modeling, when other two types of DTA knowledge (i.e., task knowledge and algorithm knowledge) have been formalized and hence can be used in existing DTA-assisted tools. The method proposed in current study focuses on DTA domain and considers the area and the terrain condition through a few simple attributes for describing the study area characteristics of a DTA application case. We also keep the similarity function on each attribute as a simpler form before more detailed research would be conducted to improve it. Preliminary evaluation results based on a case base prepared from the peer-reviewed papers we manually selected from mainstream journals of related domains show the reasonableness of the proposed case-based method. The design of both the attributes and the similarity calculation methods could be improved to reflect the domain-specific application-context knowledge more efficient, which needs additional research. For example, if the case base is with a large size, a machine learning algorithm would be available for calibrating the parameter-settings for similarity functions automatically. We will revise the manuscript to discuss the research issues in future work.

Comment 2: Presumption that articles published in good journals are supposed to provide good solutions for their specific study areas based on experts' experience and knowledge of the target task can be justified in general, but it is probably too optimistic in some cases even considering that determination of drainage network is probably only marginal problem for a part of articles. So no every solution published in good journal have to be well. And therefore a method based on selection the only one 'exemplary' published solution I feel as problematic.

Response: We agree with the referee that the solutions presented in articles published in good journal might not be optimal. In this study we assumed that those solutions are normally good for their specific study areas based on experts' experience and

knowledge of the target task. We manually selected the peer-reviewed papers related to the drainage network extraction applications which were published in mainstream journals of related domains. By this means the cases used could be kept as accurate (or reliable) as possible. Additional research is needed to enhance the proposed method by taking the reliability of the case into account. Although the solution from the case-based method might not be perfect, the method proposed in this study might automate the DTA modeling process, which makes it easy for users (especially non-expert users), and meanwhile the solution could be reasonable comparatively. This is valuable especially for non-expert users at the beginning of the modeling when field data for evaluation might be not easy to be obtained. We will revise the manuscript to include above discussion.

Comment 3: While the suggested computation of similarity of individual attributes between the new application and published one can be acceptable, the synthesis (computation of 'overall similarity') is more problematic. No (equal) weighting of used attributes is a basic problem. It is very improbable that similarity in name of target task, cell size, area, relief, slope distribution and hypsometric integral will have the same effect on determination of proper catchment area threshold for extracting drainage networks.

Response: In current method proposed, the overall similarity between a case and a new application problem is determined by applying a minimum operator to synthesizing the similarity values on every attributes in a cautious manner. In the geographical domain, a minimum operator based on the limiting factor principle is often used to synthesize similarity values on multiple attributes (Zhu and Band, 1994). This synthesis by a minimum operator means that the overall similarity result is lower (i.e., higher uncertainty for reasoning result) than it from other synthesis means such as weighted-average. Based on the experiment shown in the original manuscript, we also tested the effect of calculating the overall similarity by a simple average operator (a representative of weighted-average) instead of the minimum operator. The evaluation results show that the overall similarity for every case increased and the lowest overall similarity

among results for 50 evaluation cases increased from 0.47 to 0.68 when the minimum operator was replaced by the simple average operator. Among 50 evaluation cases, the solutions for 13 evaluation cases from the proposed method changed because the cases with the highest similarity resulted by the simple average operator were different from those resulted by the minimum operator. Due to the synthesis by the simple average operator instead of the minimum operator, the relative deviation of river density (E) increased for 10 of these 13 evaluation cases with different solutions, when E slightly decreased for other 3 evaluation cases. The increase of E even reached 20∼80 times for some cases with the overall similarity values larger than 0.8. Because the overall similarity values were larger than 0.8 for most of evaluation cases, there is no a reasonable relationship between the overall similarity value and the E. This shows that the proposed method performed poorly when the simple average operator was used instead of the minimum operator. Note that the simple average is the common representative of weighted-average, and currently it is difficult to choose a more complex weighted-average for synthesizing similarity values on multiple attributes. Therefore the synthesis by a minimum operator is proposed for current method in this study. Additional research is needed to evaluate the similarity calculation method through further test with more types of DTA applications. We will revise the manuscript to include above discussion.

Comment 4: Evaluation of experimental results is very problematic. Authors write (23-25, p.13): "Four levels of E were established empirically to reflect the reasonableness level: reasonable ([0,0.1]), acceptable ( (0.1,0.25]), questionable ( (0.25,0.5]), and unreasonable ( (0.5,+1))." It is non committal for me and if authors do not specify this 'empirical establishment' I feel it as fully subjective division. Why the difference in drainage density is unreasonable only exceed 50 %?! It smell by purpose made establishment of intervals to show "that the proposed method performs satisfactorily" (9, p.14).

Response: Four levels of E were established empirically for a summarized discussion

on the evaluation results in this study. We have realized that it is subjective to say "reasonable" based on this level of E. We will use the "deviation level" intead of "reasonableness level" to analyze the results by the solutions from the proposed method. The manuscript will be revised to avoid the misleading problem from the subjective wording for the E levels. The evaluation results (Table 3 in the manuscript) show that normally the larger the overall similarity value from the proposed method, the less is the relative deviation of river density (E). This means that the proposed method performs reasonablely.

Comment 5: The title of the paper is too complex and not quite clear. A simplification is suitable (e.g. Case-based formalization and reasoning method for digital terrain analysis – determining the catchment area threshold for extracting drainage networks).

Response: Thanks for the referee's suggestion. This manuscript proposes a case-based formalization for DTA application-context knowledge and the corresponding case-based reasoning method. The determination of catchment area threshold for extracting drainage networks was taken as an example to evaluate the proposed method. Therefore, we plan to change the title of the manuscript to be "Case-based knowledge formalization and reasoning method for digital terrain analysis âŤĂ Application to determining the catchment area threshold for extracting drainage networks".

Comment 6: Because equal weight of all attributes the binary attribute 'the name of the target task' exclude (in final comparison) all cases with another name of the target task. What is the reason of such hard limit? How was determined the attribute for particular cases? Names of types and their occurrence should be added for better understanding.

Response: Current method uses the boolean function to calculate the similarity on the nominal attribute "name of target task". This is a strict limit to prevent the proposed method from determining a case to be the solution case for a new application problem with a totally different task. In current experiment, we manually selected the
peer-reviewed papers related to the drainage network extraction applications to pre-
pare the case base. Thus all cases have same name of target task, i.e., drainage
network extraction. More detailed research on the classificiation of target task, such as
hierarchical classification or fuzzy classification, would be helpful to relax this limit on
the attribute "name of target task", which is a part of future research. We will revise the
manuscript to discuss this issue.

Comment 7: Attribute relief - is it one number for the whole area (then it very depen-
dents on area size) or average value computed by what way? (moving window - the
size and shape?)

Response: Here it means the total relief of the study area, which is the maximum
minus minimum elevation within the study area. We will revise the manuscript to use
the term "total relief" to make it clear. Two cases with similar values of total relief and
very different area sizes will have a low overall similatity from the proposed method,
because of their low similarity on the area attribute and the overall similairty calculation
by a minimum operator. Here the overall similairty calculation by a minimum operator
is more effective than that by a weighted-average operator.

Comment 8: Slope is scale dependent variable so distribution of slopes depend on grid
size. Using of cumulative frequency distribution solve this problem only partially.

Response: Yes, the slope cumulative frequency was used in this study instead of the
slope frequency distribution to describe the slope distribution attribute and relieve the
DEM resolution effect. Because of the attribute "cell size" in the case and and the
overall similairty calculation by a minimum operator, two cases with similar slope cu-
mulative frequency and very different cell sizes will have a low overall similarity from
the proposed method. We will revise the manuscript to state this point.

Comment 9: Similarity functions seem to be determined subjectively. Why difference in
magnitude of cell size (and area) can better reflect the level of similarity between DTA
applications than the numerical difference in cell size? Why is used natural log in one

case and common in another? Etc.

Response: The similarity function for each individual attribute was designed empirically to be compatible with the value type of the attribute and in accord with domain knowledge regarding the level of similarity due to the difference in the attribute value between the new application problem and an existing case. Specific to the attribute "cell size", the design of its similarity function is mainly based on two reasons. First, the numerical difference in cell size does not make sense. Taking an application with 10-m resolution as example, another application with a coarser resolution of 25 m is comparable to it from a cell size perspective, while on the other hand the resolution must be larger than 0 m. Secondly, a bell-shaped similarity function for a logarithmic transformation of cell size could balance the decrease of similarity value for those situations with a coarser resolution or a finer resolution. The similarity function for the attribute "area" is designed similarly. Because of the different characteristics of other attributes, their similarity functions are designed to be with different forms. The reason for the design of the similarity function on each attribute will be stated clearly in the revised manuscript.

Comment 10: In regard to aforementioned problems I cannot recommend the paper in the present form, (presented experiment is not enough documented to support the interpretations and conclusions). However, majority of problems could be eliminated by selection of more appropriate method of synthesis. I think, multidimensional regression is a way. This method provide for elimination of inappropriate possible influence of particular problematic published case studies (ii), reveal various weights (suitability) of used attributed and similarity functions (mainly if hierarchical partitioning will be used) (iii) and last but not least alternative results (using various attributes and methods of similarity computation) can be compared to find the most appropriate regression equation. Suitability of selected attributes and methods can be documented by this way (i) and it can partly also substitute problematic way of evaluation in this paper (iv).

Response: In current method, the overall similarity is synthesized by applying a minimum operator to the similarity values on every attributes in a cautious manner. It is based on the limiting factor principle and can prevent the proposed method from some unreasonable performance. Please also see our responses above to the seventh and eighth item of comments from anonymous referee #2. Based on the experiment shown in the original manuscript, we also tested the effect of calculating the overall similarity by a simple average operator (a representative of weighted-average) instead of the minimum operator. The expeimental results show that the proposed method with a minimum operator performs more reasonablely. Please also see our response above to the third item of comments from anonymous referee #2. We will revise the manuscript to include above discussion. Thanks for the referee's suggestion on the multidimensional regression for synthesizing individual similarity values. For a case base with large size, a machine learning algorithm would be available for calibrating the parameter-settings for similarity functions automatically. The size of case base does matter. Considerring that the size of current case based is still comparatively limited when a part of it was used as the set of indenpendent evaluation cases, we think that automatic or semi-automatic methods of creating cases should be developed to speed up the expansion of the case base (not only for the current target task, but also for other DTA application tasks). Subsequently the multidimensional regression and other machine learning methods could be tested for their effectiveness on this issue. We will revise the manuscript to discuss the research issues in future work.

Comment 11: Please also note the supplement to this comment: http://www.hydrol-earth-syst-sci-discuss.net/hess-2015-539/hess-2015-539-RC2-supplement.pdf

Response: Thanks for the referee's detailed comments marked in the original manuscript. For those on syntax errors in the original manuscript, we will revise accordingly. For other marked comments (numbered as Comment 11a∼11j below), the item-by-item responses are listed as follows.

Comment 11a: Page 3, lines 31-32. "the case-based method can simplify knowledge acquisition into case acquisition, with no need for an explicit expression model of domain knowledge" – and it is a problem.

Response: We will revise this sentence as follow to avoid misleading. Compared with traditional rule-based knowledge representation and reasoning methods, the case-based method transforms knowledge acquisition into case acquisition, with no need for an explicit expression of domain knowledge. Therefore the case-based method is suitable for DTA application-context knowledge which is non-systematic and largely tacit knowledge.

Comment 11b: Page 5, lines 25-27. "The optional output part of the case-based formalization does not currently need to be considered for the DTA domain because normally there is no change in the application context of a DTA application case when the DTA model is applied." – ?

Response: We will revise this sentence as follow to avoid confusing. The output part of a case, which is optional in the case-based formalization (Kolodner, 1993), is set to be null in this study because normally there is no change in the application context of a DTA application problem when the solution of this case is applied to this application problem.

Comment 11c: Page 6, lines 2-3. "The solution of the case with the highest similarity is reused for the new DTA application problem" – why?

Response: The case with the highest similarity means it with the most similar application context cosiderred. According to the case-based reasoning principle that solutions for similar problems are often similar, the solution of the case with the highest similarity is reused for the new DTA application problem. We will revise the manuscript to state it clearly.

Comment 11d: Page 10, lines 4-5. "the difference in magnitude of cell size can better reflect the level of similarity between DTA applications than the numerical difference in cell size." – why?

Response: Please see our response above to the 9th item (on the design of the similarity function on cell size) of comments from anonymous referee #2.

Comment 11e: Page 11, lines 21-23. That means all attributes are considered as equally significant and limiting. This assumption is not supported by any arguments.

Response: In current method, the overall similarity is synthesized by applying a minimum operator to the similarity values on every attributes in a cautious manner. It is based on the limiting factor principle and is often used to synthesize similarity values on multiple attributes in the geographical domain (Zhu and Band, 1994). We also tested the effect of calculating the overall similarity by a simple average operator (a representative of weighted-average) instead of the minimum operator. The exeprimental results show that the proposed method with a minimum operator performs more reasonabley. Please also see our response above to the third item of comments from anonymous referee #2. We will revise the manuscript to make a fruther discussion on it.

Comment 11f: Page 12, lines 26-27. "These articles are supposed to provide good solutions for their specific study areas based on experts' experience and knowledge of the target task" – really?

Response: In this study we assumed that the solutions presented in articles published in mainstream journals of related domains are normally good (might not be optimal) for their specific study areas based on experts' experience and knowledge of the target task. We manually selected the peer-reviewed papers related to the drainage network extraction applications which were published in mainstream journals of related domains. By this means the cases used could be kept as accurate (or reliable) as possible. Please also see our response above to the second item of comments from anonymous referee #2. We will revise the manuscript to avoid misleanding.

Comment 11g: Page 13, line 16. "the relative error of river density" – Is it really error? Only if we suppose a perfect settings of CA thresholds in all studies (that is unjustified presumption). Moreover, why river density and no directly CA thershold was used for

definition of the 'error'?

Response: We will revise the manuscript to use the term "relative deviation of river density" instead of the relative error of river density to avoid misleading. The deviations between the CA threshold values for different cases are highly varied (about 10-3 ~103 km2). Therefore the relative deviation of river density was used as an index for comparison between the results for different application problems and quantitative evaluation of the proposed method. We will revise the manuscript to state this point.

Comment 11h: Page 20. The similarity function on the relief attribute – ?

Response: In this study the attribute "relief" means the total relief of the study area. We will revise the manuscript to use the term "total relief" and also precisely define the calculation of the total relief, i.e., the maximum minus minimum elevation within the study area. As the description in Section 4.2.4 in the original manuscript, the similarity function for the total relief attribute was designed as a linear function using the absolute difference between the total relief of the new DTA application problem and that of existing case. Corresponding to a zero similarity value, the maximum difference between two total relief values is the larger of the total relief differences between the new application problem values and each of two extreme cases (a flat area with a total relief of zero, and an area with relief from the 8848 m of Mount Everest to sea level). So is the similarity function for the total relief attribute shown in Table 2.

Comment 11i: Page 20. The similarity function on the hypsometric curve – ?mistake

Response: Here is no mistake. The design of the similarity function for the attribute "hypsometric curve" is based on the hypsometric integral (HI). The form of the function is similar to that of the total relief attribute. The similarity on HI is 0 for the maximum possible deviation from the HI of the new application problem. So is the similarity function for this attribute shown in Table 2. Please see Section 4.2.6 in the manuscript for the description on this design.

**References**

Kolodner, J.: Case-based Reasoning, Morgan Kaufmann Publishers, San Mateo, 1993.

Zhu, A-X. and Band, L.: A knowledge-based approach to data integration for soil mapping, Can. J. Remote Sens., 20, 408-418, 1994.

---

## Author Response (AR1)

| 1        | Revisions and responses on HESS-2015-539                                                                                                               |
|----------|--------------------------------------------------------------------------------------------------------------------------------------------------------|
| 2        | ("Case-based formalization and reasoning method for                                                                                                    |
| 3        | knowledge in digital terrain analysis — Illustrated by                                                                                                 |
| 4        | determining the catchment area threshold for                                                                                                           |
| 5        | extracting drainage networks")                                                                                                                         |
| 6        |                                                                                                                                                        |
| 7        | The authors thank two anonymous referees and Dr. M. Chen for the constructive                                                                          |
| 8        | comments which are helpful for improving the final version of this manuscript. We                                                                      |
| 9        | make a point-by-point reply to these comments as below. A marked-up manuscript                                                                         |
| 10       | version showing the changes made is attached at the end of this document.                                                                              |
| 11       |                                                                                                                                                        |
| 12       | With regards to comments from the anonymous referee #1:                                                                                                |
| 13       |                                                                                                                                                        |
| 14       | Comment 1: p. 1 l. 19 DTA-assisted tools (e.g., ArcGIS, GRASS, SAGA, White Box,                                                                        |
| 15
16 | TauDEM) ArcGIS and GRASS are large, general purpose GIS packages which include DTA tools, - reference to specific modules is needed here.              |
| 17       | Response: DTA-assisted tools include general purpose GIS packages with DTA                                                                             |
| 18       | functionality (e.g., "Spatial Analyst" toolbar in ArcGIS, r.* modules in GRASS,                                                                        |
| 19       | "Terrain Analysis" menu in SAGA, etc.) and domain-specific software (e.g.,                                                                             |
| 20       | Whitebox, TauDEM, etc.) (Hengl and Reuter, 2009). We have revised the manuscript                                                                       |
| 21       | to clarify this point (the third paragraph of Section 1).                                                                                              |
| 22       |                                                                                                                                                        |
| 23
24 | Comment 2: l. 25 I find the following sentence confusing algorithm knowledge, which is the metadata of a DTA algorithm - what do authors mean by this? |

25 Response: The algorithm knowledge is the metadata of a DTA algorithm

| 1                          | (including its parameters), such as the data type of input/output file, the number of                                                                                                                                                                                                                                                                                        |  |
|----------------------------|------------------------------------------------------------------------------------------------------------------------------------------------------------------------------------------------------------------------------------------------------------------------------------------------------------------------------------------------------------------------------|--|
| 2                          | parameters, and the valid range for each parameter. We have revised the manuscript                                                                                                                                                                                                                                                                                           |  |
| 3                          | to clarify this point (the second paragraph of Section 1).                                                                                                                                                                                                                                                                                                                   |  |
| 4                          |                                                                                                                                                                                                                                                                                                                                                                              |  |
| 5
6                     | Comment 3: p.2 l. 1 again ModelBuilder is not DTA-assisted tool - not clear what is meant here                                                                                                                                                                                                                                                                               |  |
| 7                          | Response: ModelBuilder module in ArcGIS uses task knowledge and algorithm                                                                                                                                                                                                                                                                                                    |  |
| 8                          | knowledge to aid connecting a set of DTA algorithms to be an executable DTA                                                                                                                                                                                                                                                                                                  |  |
| 9                          | workflow in a interactive visual way. We have revised the manuscript to make it clear                                                                                                                                                                                                                                                                                        |  |
| 10                         | (the third paragraph of Section 1).                                                                                                                                                                                                                                                                                                                                          |  |
| 11                         |                                                                                                                                                                                                                                                                                                                                                                              |  |
| 12
13
14
15
16 | Comment 4: p.3, l. 6 this assumes that there is no validation data available - isn't
the best way to find the optimal parameters running the tools with a set of
parameters and find the best fit with the field data (or remotely sensed data if they
provide sufficient information)? What if the case studies are inaccurate? Can this
be taken into account? |  |
| 17                         | Response: We agree with the reviewer that the best way to determine the optimal                                                                                                                                                                                                                                                                                              |  |
| 18                         | parameter-settings should be the evaluation based on the field data. However, at the                                                                                                                                                                                                                                                                                         |  |
| 19                         | beginning of the modeling, field data might be not easy to be obtained and the                                                                                                                                                                                                                                                                                               |  |
| 20                         | evaluation process is not easy to operate for those non-expert users. The method                                                                                                                                                                                                                                                                                             |  |
| 21                         | proposed in this study might automate the DTA modeling process, which makes it                                                                                                                                                                                                                                                                                               |  |
| 22                         | easy for users (especially non-expert users), and meanwhile the result model could be                                                                                                                                                                                                                                                                                        |  |
| 23                         | reasonable comparatively. We have revised the manuscript to discuss this point (the                                                                                                                                                                                                                                                                                          |  |
| 24                         | second paragraph of Section 1, and the first paragraph of Section 6).                                                                                                                                                                                                                                                                                                        |  |
| 25                         | The algorithm and parameter-settings presented in those journal papers might not                                                                                                                                                                                                                                                                                             |  |
| 26                         | be optimal, thus the corresponding cases might be inaccurate. In this study, we                                                                                                                                                                                                                                                                                              |  |

| applications which were published in mainstream journals of related domains. By this means the cases used could be kept as accurate (or reliable) as possible. Additional research is needed to enhance the proposed method by taking the reliability of the case into account. We have revised the manuscript to discuss this point (the first paragraph of Section 5.1.1, and the second paragraph of Section 6). Comment 5: p. 7 l. 15 What is meant by aspect here? Response: Here the "aspect" means the kind of attributes designed to describe the terrain condition. We have revised the manuscript to use the term unambiguously. |
|---------------------------------------------------------------------------------------------------------------------------------------------------------------------------------------------------------------------------------------------------------------------------------------------------------------------------------------------------------------------------------------------------------------------------------------------------------------------------------------------------------------------------------------------------------------------------------------------------------------------------------------------------|
| means the cases used could be kept as accurate (or reliable) as possible. Additional research is needed to enhance the proposed method by taking the reliability of the case into account. We have revised the manuscript to discuss this point (the first paragraph of Section 5.1.1, and the second paragraph of Section 6).
Comment 5: p. 7 l. 15 What is meant by aspect here?
Response: Here the "aspect" means the kind of attributes designed to describe the terrain condition. We have revised the manuscript to use the term unambiguously.                                                                                |
| research is needed to enhance the proposed method by taking the reliability of the case into account. We have revised the manuscript to discuss this point (the first paragraph of Section 5.1.1, and the second paragraph of Section 6). Comment 5: p. 7 l. 15 What is meant by aspect here? Response: Here the "aspect" means the kind of attributes designed to describe the terrain condition. We have revised the manuscript to use the term unambiguously.                                                                                                                                                                           |
|  <li>case into account. We have revised the manuscript to discuss this point (the first paragraph of Section 5.1.1, and the second paragraph of Section 6).</li> <li>Comment 5: p. 7 l. 15 What is meant by aspect here?</li> <li>Response: Here the "aspect" means the kind of attributes designed to describe the terrain condition. We have revised the manuscript to use the term unambiguously.</li>                                                                                                                                                                                                                         |
| <pre>paragraph of Section 5.1.1, and the second paragraph of Section 6).</pre> Comment 5: p. 7 l. 15 What is meant by aspect here? Response: Here the "aspect" means the kind of attributes designed to describe the terrain condition. We have revised the manuscript to use the term unambiguously.                                                                                                                                                                                                                                                                                                                                             |
| Comment 5: p. 7 l. 15 What is meant by aspect here?
Response: Here the "aspect" means the kind of attributes designed to describe the terrain condition. We have revised the manuscript to use the term unambiguously.                                                                                                                                                                                                                                                                                                                                                                                                                  |
| Comment 5: p. 7 l. 15 What is meant by aspect here?
Response: Here the "aspect" means the kind of attributes designed to describe the terrain condition. We have revised the manuscript to use the term unambiguously.                                                                                                                                                                                                                                                                                                                                                                                                                  |
| Response: Here the "aspect" means the kind of attributes designed to describe the terrain condition. We have revised the manuscript to use the term unambiguously.                                                                                                                                                                                                                                                                                                                                                                                                                                                                                |
| terrain condition. We have revised the manuscript to use the term unambiguously.                                                                                                                                                                                                                                                                                                                                                                                                                                                                                                                                                                  |
|                                                                                                                                                                                                                                                                                                                                                                                                                                                                                                                                                                                                                                                   |
|                                                                                                                                                                                                                                                                                                                                                                                                                                                                                                                                                                                                                                                   |
| Comment 6: l. 17 how do you compute relief - you refer to it as steep or gently sloping - isn't that equivalent to slope? Relief in gemorphometry is a very specific metrics - specify here what you are using or use different term                                                                                                                                                                                                                                                                                                                                                                                                              |
| Response: Here it means the total relief of the study area, which is the maximum                                                                                                                                                                                                                                                                                                                                                                                                                                                                                                                                                                  |
| minus minimum elevation within the study area. We have revised the manuscript to                                                                                                                                                                                                                                                                                                                                                                                                                                                                                                                                                                  |
| use the term properly.                                                                                                                                                                                                                                                                                                                                                                                                                                                                                                                                                                                                                            |
|                                                                                                                                                                                                                                                                                                                                                                                                                                                                                                                                                                                                                                                   |
| Comment 7: l. 24 seven grades? did you meant seven classes or categories? It appears that you mix relief and slope - perhaps use equations to precisely define what you mean                                                                                                                                                                                                                                                                                                                                                                                                                                                                      |
| Response: Yes, the slope gradient value was divided into seven classes. We have                                                                                                                                                                                                                                                                                                                                                                                                                                                                                                                                                                   |
| revised the manuscript to make it clear and also precisely define the calculation of the                                                                                                                                                                                                                                                                                                                                                                                                                                                                                                                                                          |
| total relief used in this study (the fourth paragraph of Section 4.1).                                                                                                                                                                                                                                                                                                                                                                                                                                                                                                                                                                            |
|                                                                                                                                                                                                                                                                                                                                                                                                                                                                                                                                                                                                                                                   |
|                                                                                                                                                                                                                                                                                                                                                                                                                                                                                                                                                                                                                                                   |
|                                                                                                                                                                                                                                                                                                                                                                                                                                                                                                                                                                                                                                                   |

**1 classes?**

Response: Yes, we have revised the manuscript to make it clear and use the term
"elevation-slope cumulative frequency distribution" instead of the "relief-slope
cumulative frequency distribution" used in the original manuscript. The legend of Fig.
3 has been revised accordingly.

6

**7 Comment 9: l. 30 relieves the DEM resolution effect ? what do you mean by 8 relieves?**

9 Response: DEM resolution has a strong influence on calculating the slope gradient and its frequency distribution (Chang and Tsai, 1991; Grohmann, 2015), while the 10 DEM resolution has a comparatively weak influence on the cumulative frequency 11 distribution of slope gradient. To relieve this DEM resolution effect and ensure the 12 comparability of slope distributions from two cases with different DEM resolutions, 13 we used the slope cumulative frequency in this study instead of the slope frequency 14 distribution to describe the slope distribution. We have revised the manuscript to 15 clarify this point (the fifth paragraph of Section 4.1). 16

17

18 Comment 10: p. 8 l. 20 comment - environmental conditions, especially the 19 groundwater level could be more important than the topo parameters, so the case 20 studies used should be evaluated for this and those where parameters other than the 21 proposed ones play determining role should be excluded

Response: We agree with the reviewer that the groundwater level also plays important role on drainage network formation. However, the information of groundwater level is often difficult to be collected. Normal way of drainage network extraction by DTA is mainly based on topographic information. The method proposed in current study focuses on DTA domain and considers the area and the terrain condition for describing the study area characteristics of a DTA application case.
Preliminary evaluation shows the reasonableness of the proposed method. The design
of the attributes used to describe the problem part of a case could be improved to
describe the domain-specific application-context information in a more adaptive and
efficient manner, which needs additional research. We have revised the manuscript to
discuss this issue (the second paragraph of Section 6).

7

**8 Comment 11: p. 9 l. 17 - Doesn't the need to empirically adjust the shape of the bell 9 curve beat the purpose of the proposed method?**

10 Response: Currently we empirically set the shape-adjusting parameter (w) with fixed values for two attributes with bell-shaped similarity function. Preliminary 11 evaluation shows that the proposed method with these settings performs well. The 12 13 way of setting the shape-adjusting parameter will be explored as a part of future research. For example, if case base is with a large size, a machine learning algorithm 14 might be available for calibrating the shape-adjusting parameter automatically. We 15 16 have revised the manuscript to include the discussion on this issue (the first paragraph of Section 4.2, and the third paragraph of Section 6). 17

18

**19 Comment 12: eq. 1 ln(0.5) is a constant - why ln and not the constant value 20 directly?**

Response: We accepted this advice and have revised Eq. (1) accordingly.

21

22

Comment 13: p. 10 l. 4 and 5 magnitude of cell size - did you mean absolute value? Magnitude does not make sense here. If it is indeed absolute value (as indicated in Table 2), this treats cell size larger the same as cell size smaller - there is a fundamental difference between downscaling and upscaling or going to higher

**level of detail versus lower level of detail in terms of stream extraction - how do you account for this issue?**

Response: In this study, we try to keep the similarity function on each attribute as 3 a simpler form before more detailed research could be conducted to improve it. 4 Current design of the similarity function for cell size is mainly based on two reasons. 5 First, the numerical difference in cell size does not work. Taking an application with 6 10-m resolution as example, another application with a coarser resolution of 25 m is 7 comparable to it from a cell size perspective, while on the other hand the resolution 8 cannot be less than or equal to 0 m. Secondly, a bell-shaped similarity function for a 9 10 logarithmic transformation of cell size could balance the decrease of similarity value for those situations with a coarser resolution or a finer resolution. Note that the 11 similarity value on cell size will rapidly decrease to be about 0.58 when the resolution 12 13 is coarsened to be double the resolution of a case or is refined to be a half of the case's resolution. The lower similarity value will deny the corresponding case to be a 14 credible solution provider for the new application problem. This means that the 15 16 proposed method does not suggest a large-step downscaling and upscaling application of existing cases. We have revised the manuscript to state this point (Section 4.2.2). 17

18

**Comment 15: p. 10 l. 15 - what is meant by area - total area of the study site? magnitude here probably should be again the absolute value**

Response: Yes, the area attribute is the total area of the study site. In this study we design a bell-shaped similarity function for a logarithmic transformation of area based on the idea similar to the design for the cell size attribute. Please also see our response above to the 14th item of comments from anonymous referee #1. We have revised the 1 manuscript to make this point clear (Section 4.2.3).

2

Comment 16: p. 10 l. 22 it is not clear what is meant by relief here - providing an equation or more precise definition is necessary, is it the difference between the minimum and maximum elevation in the study area? If yes, please check how the term relief is used in literature and what should you use here.

7 Response: We have revised the manuscript to use the term "total relief" and also

8 precisely define the calculation of the total relief, i.e., the maximum minus minimum

9 elevation within the study area (the fourth paragraph of Section 4.1).

10

| 11
12
13
14
15
16 | Comment 17: p. 12 l. 1 the presented workflow applies to only the older algorithms
and is highly simplified - this needs to be mentioned. For example, filling of pits
(many are often real) and flat areas is not necessary if least cost path algorithm is
used - see e.g. Metz et al. 2011, doi:10.5194/hess-15-667-2011r the second step also
is not quite accurate - spatial distribution of catchment area sounds confusing -
perhaps you meant flow accumulation or contributing areas for each grid cell? |
|----------------------------------|----------------------------------------------------------------------------------------------------------------------------------------------------------------------------------------------------------------------------------------------------------------------------------------------------------------------------------------------------------------------------------------------------------------------------------------------------------------------------------------------------------------------------------|
| 17                               | Response: We accepted this advice and have revised the manuscript accordingly                                                                                                                                                                                                                                                                                                                                                                                                                                                    |
| 18                               | (the first paragraph of Section 5.1). A new reference (Metz, M., Mitasova, H., and                                                                                                                                                                                                                                                                                                                                                                                                                                               |
| 19                               | Harmon, R. S.: Efficient extraction of drainage networks from massive, radar-based                                                                                                                                                                                                                                                                                                                                                                                                                                               |
| 20                               | elevation models with least cost path search, Hydrol. Earth Syst. Sci., 15, 667-678,                                                                                                                                                                                                                                                                                                                                                                                                                                             |
| 21                               | 2011) have been cited in the revised manuscript.                                                                                                                                                                                                                                                                                                                                                                                                                                                                                 |
|                                  |                                                                                                                                                                                                                                                                                                                                                                                                                                                                                                                                  |

22

Comment 18: l. 15 it is apparent that the proposed experiment applies only to ArcGIS-based workflow which is highly limited and somewhat obsolete, but it can still be used as a case study, given the large number of users who would use this tool. Were all the articles used as case base using the same algorithm?

27 Response: In most of articles used for case preparation a single flow direction
28 algorithm (such as D8 algorithm) was adopted, when a few articles did not state
29 clearly the flow direction algorithm used. Note that the experiment in this study was

designed to focus on the determination of CA threshold for drainage network
extraction, not the flow direction algorithm used. We have revised the manuscript to
state this point (the first paragraph of Section 5.1.1).

4

**5 Comment 19: p. 12 l. 29, 30 - what is meant by extracting here? perhaps 6 identifying?**

Response: The main work involved in preparing the case problem was to specify
each attribute of the study area according to the article. We have revised the
manuscript accordingly.

10

**11 Comment 20: did all articles use SRTM or ASTER?**

Response: Some articles used for case preparation in this study used DEM with a finer resolution than that of SRTM DEM or ASTER GDEM. However, those DEM are often not easy to collect by us. Therefore, we used these open DEM data to derive the case attributes such as area, total relief, elevation-slope cumulative frequency distribution, and hypsometric curve. And this process also makes each of these attributes comparable between different cases. We have stated this point in the revised manuscript (the second paragraph of Section 5.1.1).

19

**Comment 21: It is not clear why river density for evaluations - how is it computed? Why not the total length of the river network? How many validation cases lead to shorter streams and how many were longer (see Fig. 4).**

Response: The river density was calculated by the total length of the extracted drainage network divided by the area of the study site. In current manuscript, the relative deviation of river density was used as an index for quantitative evaluation of

the proposed method. Based on Eq. (2) in the manuscript, which defines this index,
the index value will be same if the total length of river network is used instead of the
river density. Compared with the length of drainage network, the river density can
also be used to make comparison between the results for different application
problems, although this comparison has not been made for discussion in current
manuscript.

The counts of validation cases which got shorter and longer drainage networks
from the proposed method are 16 and 28, respectively. We have revised the
manuscript to provide this information (the first paragraph of Section 5.2).

10

**11 With regards to comments from the anonymous referee #2:**

12

Comment 1: Authors suggest to replace a deep functional analysis of application context by the method based on learning from various previous solutions regardless of their detailed knowledge. OK, deeper functional analysis can be too difficult and selection of only some elements of application context can be a solution. However selection of used attributes and similarity functions was reasoned only poorly and in no way verified.

Response: In this study we explored how to formalize application-context knowledge in DTA and apply it to DTA modeling, when other two types of DTA knowledge (i.e., task knowledge and algorithm knowledge) have been formalized and hence can be used in existing DTA-assisted tools. The method proposed in current study focuses on DTA domain and considers the area and the terrain condition through a few simple attributes for describing the study area characteristics of a DTA application case. We also keep the similarity function on each attribute as a simpler

form before more detailed research would be conducted to improve it. Preliminary 1 2 evaluation based on a case base prepared from the peer-reviewed papers we manually 3 selected from mainstream journals of related domains shows the reasonableness of the proposed case-based method. The design of both the attributes and the similarity 4 calculation methods could be improved to reflect the domain-specific 5 application-context knowledge more efficient, which needs additional research. For 6 example, if the case base is with a large size, a machine learning algorithm would be 7 available for calibrating the parameter-settings for similarity functions automatically. 8 9 We have revised the manuscript to discuss the research issues in future work (the second and third paragraph of Section 6). 10

11

Comment 2: Presumption that articles published in good journals are supposed to provide good solutions for their specific study areas based on experts' experience and knowledge of the target task can be justified in general, but it is probably too optimistic in some cases even considering that determination of drainage network is probably only marginal problem for a part of articles. So no every solution published in good journal have to be well. And therefore a method based on selection the only one 'exemplary' published solution I feel as problematic.

Response: We agree with the referee that the solutions presented in articles 19 published in good journal might not be optimal. In this study we assumed that those 20 21 solutions are normally good for their specific study areas based on experts' experience and knowledge of the target task. We manually selected the peer-reviewed papers 22 related to the drainage network extraction applications which were published in 23 mainstream journals of related domains. By this means the cases used could be kept 24 25 as accurate (or reliable) as possible. Additional research is needed to enhance the 26 proposed method by taking the reliability of the case into account.

Although the solution from the case-based method might not be perfect, the method proposed in this study might automate the DTA modeling process, which makes it easy for users (especially non-expert users), and meanwhile the solution could be reasonable comparatively. This is valuable especially for non-expert users at the beginning of the modeling when field data for evaluation might be not easy to be obtained.

We have revised the manuscript to include above discussion (the first paragraph of
Section 5.1.1, and the first paragraph of Section 6).

9

Comment 3: While the suggested computation of similarity of individual attributes between the new application and published one can be acceptable, the synthesis (computation of 'overall similarity') is more problematic. No (equal) weighting of used attributes is a basic problem. It is very improbable that similarity in name of target task, cell size, area, relief, slope distribution and hypsometric integral will have the same effect on determination of proper catchment area threshold for extracting drainage networks.

Response: In current method proposed, the overall similarity between a case and a 17 new application problem is determined by applying a minimum operator to 18 synthesizing the similarity values on every attributes in a cautious manner. In the 19 geographical domain, a minimum operator based on the limiting factor principle is 20 often used to synthesize similarity values on multiple attributes (Zhu and Band, 1994). 21 This synthesis by a minimum operator means that the overall similarity result is lower 22 (i.e., higher uncertainty for reasoning result) than it from other synthesis means such 23 as weighted-average. We have revised the manuscript to add a new section to state 24 25 this point (Section 4.3).

26

Based on the experiment shown in the original manuscript, we also tested the

effect of calculating the overall similarity by a simple average operator (a 1 representative of weighted-average) instead of the minimum operator. Table 3 has 2 3 been extended to include the evaluation results. The evaluation shows that the overall similarity for every case increased and the lowest overall similarity among results for 4 50 evaluation cases increased from 0.47 to 0.68 when the minimum operator was 5 replaced by the simple average operator. Among 50 evaluation cases, the solutions for 6 13 evaluation cases from the proposed method changed because the cases with the 7 highest similarity resulted by the simple average operator were different from those 8 9 resulted by the minimum operator. Due to the synthesis by the simple average operator instead of the minimum operator, the relative deviation of river density (E)10 increased for 10 of these 13 evaluation cases with different solutions, when E slightly 11 12 decreased for other 3 evaluation cases. The increase of E even reached 20~80 times for some cases with the overall similarity values larger than 0.8. Because the overall 13 similarity values were larger than 0.8 for most of evaluation cases, there is no a 14 15 reasonable relationship between the overall similarity value and the E. This shows 16 that the proposed method performed poorly when the simple average operator was used instead of the minimum operator. 17

Note that the simple average is the common representative of weighted-average, and currently it is difficult to choose a more complex weighted-average for synthesizing similarity values on multiple attributes. Therefore the synthesis by a minimum operator is proposed for current method in this study. Additional research is needed to evaluate the similarity calculation method through further test with more

types of DTA applications. 1

2 We have revised the manuscript to include above evaluation and discussion (the

3 last paragraph of Section 5.1.2, the last paragraph of Section 5.2, and Table 3).

4

Comment 4: Evaluation of experimental results is very problematic. Authors write 5 (23-25, p.13): "Four levels of E were established empirically to reflect the 6 7 reasonableness level: reasonable ([0,0.1]), acceptable ( (0.1,0.25]), questionable ((0.25,0.5]), and unreasonable ((0.5,+1))." It is non committal for me and if 8 9 authors do not specify this 'empirical establishment' I feel it as fully subjective division. Why the difference in drainage density is unreasonable only exceed 50 %?! 10 It smell by purpose made establishment of intervals to show "that the proposed 11 method performs satisfactorily" (9, p.14). 12

13 Response: Four levels of the relative deviation of river density (E) were established empirically for a summarized discussion on the evaluation results in this 14 study. We have realized that it is subjective to say "reasonable" based on this level of 15 16 E. We have revised the manuscript to use the "deviation level" intead of "reasonableness level" to analyze the results by the solutions from the proposed 17 method. The manuscript have been revised to avoid the misleading problem from the 18 19 subjective wording for the E levels (the second paragraph of Section 5.1.2). The 20 evaluation results (Table 3 in the manuscript) show that normally the larger the overall similarity value from the proposed method, the less is the E. This means that 21 the proposed method performs reasonablely. 22

23

Comment 5: The title of the paper is too complex and not quite clear. A 24 simplification is suitable (e.g. Case-based formalization and reasoning method for 25 26 digital terrain analysis – determining the catchment area threshold for extracting 27 drainage networks).

Response: Thanks for the referee's suggestion. This manuscript proposes a

1 case-based formalization for DTA application-context knowledge and the
2 corresponding case-based reasoning method. The determination of catchment area
3 threshold for extracting drainage networks was taken as an example to evaluate the
4 proposed method. Therefore, we changed the title of the manuscript to be "Case-based
5 knowledge formalization and reasoning method for digital terrain analysis —
6 Application to determining the catchment area threshold for extracting drainage
7 networks".

8

9 Comment 6: Because equal weight of all attributes the binary attribute ' the name 10 of the target task' exclude (in final comparison) all cases with another name of the 11 target task. What is the reason of such hard limit? How was determined the 12 attribute for particular cases? Names of types and their occurrence should be added 13 for better understanding.

14 Response: Current method uses the boolean function to calculate the similarity on the nominal attribute "name of target task". This is a strict limit to prevent the 15 proposed method from determining a case to be the solution case for a new 16 application problem with a totally different task. In current experiment, we manually 17 selected the peer-reviewed papers related to the drainage network extraction 18 applications to prepare the case base. Thus all cases have same name of target task, 19 i.e., drainage network extraction. More detailed research on the classificiation of 20 target task, such as hierarchical classification or fuzzy classification, would be helpful 21 to relax this limit on the attribute "name of target task", which is a part of future 22 research. We have revised the manuscript to discuss this issue (the second paragraph 23 24 of Section 4.2.1).

Comment 7: Attribute relief - is it one number for the whole area (then it very
dependents on area size) or average value computed by what way? (moving window
the size and shape?)

4 Response: Here it means the total relief of the study area, which is the maximum minus minimum elevation within the study area. We have revised the manuscript to 5 use the term "total relief" to make it clear (Section 4.2.4 and other places using this 6 term in the maniscript). Two cases with similar values of total relief and very different 7 area sizes will have a low overall similarity from the proposed method, because of 8 their low similarity on the area attribute and the overall similairty calculation by a 9 minimum operator. Here the overall similarity calculation by a minimum operator is 10 more effective than that by a weighted-average operator. We have revised the 11 manuscript to discuss this point (Section 4.3). 12

13

**14 Comment 8: Slope is scale dependent variable so distribution of slopes depend on 15 grid size. Using of cumulative frequency distribution solve this problem only 16 partially.**

Response: Yes, the slope cumulative frequency was used in this study instead of the slope frequency distribution to describe the slope distribution attribute and relieve the DEM resolution effect. Because of the attribute "cell size" in the case and and the overall similairty calculation by a minimum operator, two cases with similar slope cumulative frequency and very different cell sizes will have a low overall similarity from the proposed method. We have revised the manuscript to state this point (Section 4.3).

24

Comment 9: Similarity functions seem to be determined subjectively. Why difference in magnitude of cell size (and area) can better reflect the level of similarity between DTA applications than the numerical difference in cell size?
 Why is used natural log in one case and common in another? Etc.

Response: The similarity function for each individual attribute was designed 3 empirically to be compatible with the value type of the attribute and in accord with 4 domain knowledge regarding the level of similarity due to the difference in the 5 attribute value between the new application problem and an existing case. Specific to 6 the attribute "cell size", the design of its similarity function is mainly based on two 7 reasons. First, the numerical difference in cell size does not make sense. Taking an 8 application with 10-m resolution as example, another application with a coarser 9 10 resolution of 25 m is comparable to it from a cell size perspective, while on the other hand the resolution must be larger than 0 m. Secondly, a bell-shaped similarity 11 function for a logarithmic transformation of cell size could balance the decrease of 12 13 similarity value for those situations with a coarser resolution or a finer resolution. The similarity function for the attribute "area" is designed similarly. Because of the 14 different characteristics of other attributes, their similarity functions are designed to 15 16 be with different forms. The reason for the design of the similarity function on each attribute have been stated clearly in the revised manuscript (Section 4.2.2 and other 17 section on the design of similarity function). 18

19

Comment 10: In regard to aforementioned problems I cannot recommend the paper 20 in the present form, (presented experiment is not enough documented to support the 21 interpretations and conclusions). However, majority of problems could be 22 eliminated by selection of more appropriate method of synthesis. I think, 23 24 multidimensional regression is a way. This method provide for elimination of 25 inappropriate possible influence of particular problematic published case studies (ii), reveal various weights (suitability) of used attributed and similarity functions 26 (mainly if hierarchical partitioning will be used) (iii) and last but not least 27

alternative results (using various attributes and methods of similarity computation) 1 can be compared to find the most appropriate regression equation. Suitability of 2 selected attributes and methods can be documented by this way (i) and it can partly 3 also substitute problematic way of evaluation in this paper (iv).

4

5 Response: In current method, the overall similarity is synthesized by applying a minimum operator to the similarity values on every attributes in a cautious manner. It 6 7 is based on the limiting factor principle and can prevent the proposed method from some unreasonable performance. Please also see our responses above to the seventh 8 and eighth item of comments from anonymous referee #2. Based on the experiment 9 shown in the original manuscript, we also tested the effect of calculating the overall 10 11 similarity by a simple average operator (a representative of weighted-average) instead of the minimum operator. The experimental results show that the proposed method 12 with a minimum operator performs more reasonablely (the last paragraph of Section 13 14 5.2; Table 3 has been extended to include the new experimental resuls). Please also see our response above to the third item of comments from anonymous referee #2. We 15 have revised the manuscript to include above discussion (Section 4.3, the last 16 17 paragraph of Section 5.1.2, and the last paragraph of Section 5.2 in the revised manuscript). The discussion on the experimental results (the first two paragraphs of 18 Section 5.2) has been reorganized to fluently combine the extended experimental 19 results and discussion. The order of Table 4,5, and 6 in the original manuscript have 20 21 been adjusted to be Table 5, 6, and 4 in the revised manuscript.

Thanks for the referee's suggestion on the multidimensional regression for 22 23 synthesizing individual similarity values. For a case base with large size, a machine learning algorithm would be available for calibrating the parameter-settings for 24

| 1              | similarity functions automatically. The size of case base does matter. Considerring                                                                             |
|----------------|-----------------------------------------------------------------------------------------------------------------------------------------------------------------|
| 2              | that the size of current case base is still comparatively limited when a part of it was                                                                         |
| 3              | used as the set of independent evaluation cases, we think that automatic or                                                                                     |
| 4              | semi-automatic methods of creating cases should be developed to speed up the                                                                                    |
| 5              | expansion of the case base (not only for the current target task, but also for other DTA                                                                        |
| 6              | application tasks). Subsequently the multidimensional regression and other machine                                                                              |
| 7              | learning methods could be tested for their effectiveness on this issue. We have revised                                                                         |
| 8              | the manuscript to discuss these research issues in future work (the third paragraphs of                                                                         |
| 9              | Section 6).                                                                                                                                                     |
| 10             |                                                                                                                                                                 |
| 11
12
13 | Comment 11: Please also note the supplement to this comment:
http://www.hydrol-earth-syst-sci-discuss.net/hess-2015-539/hess-2015-539-RC2-sup
plement.pdf |
| 14             | Response: Thanks for the referee's detailed comments marked in the original                                                                                     |
| 15             | manuscript. For those on syntax errors in the original manuscript, we have revised                                                                              |
| 16             | accordingly. For other marked comments (numbered as Comment 11a~11j below),                                                                                     |
| 17             | the item-by-item responses are listed as follows.                                                                                                               |
|                |                                                                                                                                                                 |
| 18             |                                                                                                                                                                 |

**acquisition into case acquisition, with no need for an explicit expression model of domain knowledge" – and it is a problem.**

Response: We have revised this sentence as follow to avoid misleading (the second paragraph of Section 2). Compared with traditional rule-based knowledge representation and reasoning methods, the case-based method transforms knowledge acquisition into case acquisition, with no need for an explicit expression of domain

- 1 knowledge. Therefore the case-based method is suitable for DTA application-context

knowledge which is non-systematic and largely tacit knowledge.

3

2

Comment 11b: Page 5, lines 25-27. "The optional output part of the case-based
formalization does not currently need to be considered for the DTA domain because
normally there is no change in the application context of a DTA application case
when the DTA model is applied." -- ?

8 Response: We have revised this sentence as follow to avoid confusing (the last 9 paragraph of Section 3.1). The output part of a case, which is optional in the 10 case-based formalization (Kolodner, 1993), is set to be null in this study because

- 11 normally there is no change in the application context of a DTA application problem
- 12 when the solution of this case is applied to this application problem.
- 13

**14 Comment 11c: Page 6, lines 2-3. "The solution of the case with the highest 15 similarity is reused for the new DTA application problem" – why?**

Response: The case with the highest similarity means it with the most similar application context considerred. According to the case-based reasoning principle that solutions for similar problems are often similar, the solution of the case with the highest similarity is reused for the new DTA application problem. We have revised the manuscript to state it clearly (the first paragraph of Section 3.2).

21

**Comment 11d: Page 10, lines 4-5. "the difference in magnitude of cell size can better reflect the level of similarity between DTA applications than the numerical difference in cell size." – why?**

25 Response: Please see our response above to the 9th item (on the design of the

- similarity function on cell size) of comments from anonymous referee #2.
- 27

Comment 11e: Page 11, lines 21-23. That means all attributes are considered as
 equally significant and limiting. This assumption is not supported by any
 arguments.

Response: In current method, the overall similarity is synthesized by applying a 4 minimum operator to the similarity values on every attributes in a cautious manner. It 5 is based on the limiting factor principle and is often used to synthesize similarity 6 values on multiple attributes in the geographical domain (Zhu and Band, 1994). We 7 also tested the effect of calculating the overall similarity by a simple average operator 8 (a representative of weighted-average) instead of the minimum operator. The 9 expeirmental results show that the proposed method with a minimum operator 10 performs more reasonablely. We have revised the manuscript to make a fruther 11 discussion on it. Please also see our response above to the third item of comments 12 from anonymous referee #2. 13

14

**Comment 11f: Page 12, lines 26-27. "These articles are supposed to provide good solutions for their specific study areas based on experts' experience and knowledge of the target task" – really?**

18 Response: In this study we assumed that the solutions presented in articles published in mainstream journals of related domains are normally good (might not be 19 optimal) for their specific study areas based on experts' experience and knowledge of 20 the target task. We manually selected the peer-reviewed papers related to the drainage 21 22 network extraction applications which were published in mainstream journals of related domains. By this means the cases used could be kept as accurate (or reliable) 23 24 as possible. We have revised the manuscript to avoid misleanding. Please also see our response above to the second item of comments from anonymous referee #2. 25

Comment 11g: Page 13, line 16. "the relative error of river density" -- Is it really
error? Only if we suppose a perfect settings of CA thresholds in all studies (that is
unjustified presumption). Moreover, why river density and no directly CA thershold
was used for definition of the 'error'?

Response: We have revised the manuscript to use the term "relative deviation of
river density" instead of the relative error of river density to avoid misleading (the
second paragraph of Section 5.1.2, and other places using this term in the revised
manuscript).

10 The deviations between the CA threshold values for different cases are highly 11 varied (about  $10^{-3} \sim 10^3 \text{ km}^2$ ). Therefore the relative deviation of river density was 12 used as an index for comparison between the results for different application problems 13 and quantitative evaluation of the proposed method. We have revised the manuscript 14 to state this point (the first paragraph of Section 5.1.1).

15

**16 Comment 11h: Page 20. The similarity function on the relief attribute --?**

Response: In this study the attribute "relief" means the total relief of the study 17 area. We have revised the manuscript to use the term "total relief" and also precisely 18 define the calculation of the total relief, i.e., the maximum minus minimum elevation 19 20 within the study area. As the description in Section 4.2.4 in the manuscript, the similarity function for the total relief attribute was designed as a linear function using 21 the absolute difference between the total relief of the new DTA application problem 22 and that of existing case. Corresponding to a zero similarity value, the maximum 23 24 difference between two total relief values is the larger of the total relief differences 25 between the new application problem values and each of two extreme cases (a flat area with a total relief of zero, and an area with relief from the 8848 m of Mount
 Everest to sea level). So is the similarity function for the total relief attribute shown in
 Table 2.

4

**5 Comment 11i: Page 20. The similarity function on the hypsometric curve 6 --?mistake**

Response: Here is no mistake. The design of the similarity function for the attribute "hypsometric curve" is based on the hypsometric integral (HI). The form of the function is similar to that of the total relief attribute. The similarity on HI is 0 for the maximum possible deviation from the HI of the new application problem. So is the similarity function for this attribute shown in Table 2. Please see Section 4.2.6 in the manuscript for the description on this design.

13

**14 With regards to comments from Dr. M. Chen:**

15 Comment 1: In your paper, as determining the CA is a simple function related to 16 DTA and DTA is also just a part of modeling method, how to deal with the 17 complexity problem when conducting comprehensive research and analysis, i.e., 18 how to formalized the complex knowledge about some complex problems, the 19 semantic problem, the structure to represent the knowledge, ect. Do you have some 20 preliminary ideas? I think this is the key step to promote your idea into application 21 in a broader field.

22 Response: This study explores how to formalize application-context knowledge in

23 DTA and apply it to DTA modeling, when other two types of DTA knowledge (i.e.,

task knowledge and algorithm knowledge) have been formalized by means of rule or

semantic networks (Russell and Norvig, 2009) and hence can be used in existing

26 DTA-assisted tools. Combining the propose method with existing methods for using

27 other two types of DTA knowledge, automated DTA modeling could be implemented

to make DTA easy to use for users (especially non-expert users) and ensure that the
result model is reasonable comparatively.

3 For other geographic modeling domains, normally the modeling knowledge could also be classified into these three types, i.e., task knowledge, algorithm knowledge, 4 and application-context knowledge. The task and algorithm knowledge in some 5 domains (e.g., watershed modeling) which are more complex than those in DTA have 6 been explored for formalization and inference methods and corresponding tools, such 7 as Gregersen et al. (2007) and Škerjanec et al. (2014) in automated watershed 8 9 modeling domain. For those geographic modeling domains in which the application context knowledge is also largely non-systematic and tacit knowledge, the case-based 10 idea proposed in this manuscript could also be available to combining with the 11 12 existing automated modeling methods of using the task and algorithm knowledge in these domains. We have revised the manuscript to include the discussion on this issue 13 (the first and the last paragraphs of Section 6). Three new references (Gregersen et al., 14 15 2007; Škerjanec et al., 2014; Lin et al., 2013) have been added and cited in the revised manuscript for the discussion. 16

17

Comment 2: In page 2, you mentioned that "However, current DTA-assisted tools...provide very limited support during the DTA application modeling process".
The conclusion is somewhat arbitrary, you may need to provide more arguments here.

Response: Currently, there is no well-established formalization method for application-context knowledge. Existing DTA-assisted tools, which have used the task knowledge and algorithm knowledge, consequently cannot use this type of

knowledge to provide more effective support to DTA application modeling process.
This situation exists mainly because this type of DTA knowledge is largely
non-systematic and tacit knowledge, and often exists only in documents for specific
case studies (DTA application instances) or even just in the experience of domain
experts. We have revised the manuscript to state this point (the fourth paragraph of
Section 1).

7

**8 Comment 3: page 3, line 6, "largely inaccurate"**

9 Response: The application-context knowledge of DTA is is largely non-systematic
10 and tacit knowledge. We have revised the manuscript accordingly (the fourth
11 paragraph of Section 1).

12

**13 *Comment 4: page 4, line 19, "is not necessary to participate", why? please explain it* 14 *clearly.**

Response: Only the problem part of each case is used to calculate the similarity between the case and the new application problem. The solution of the case with the highest similarity is retrieved as the solution for the new DTA application problem. Thus the solution part of a case does not participate in the reasoning procedure. We have revised the manuscript to state this point (the first paragraph of Section 3.1).

20

**Comment 5: Page 7, part 4.1, I think maybe you need to provide a table here to explain your quantitative attributes, not just some sentences.**

- Response: Table 2 lists the attributes used to formalize a case problem and the
- corresponding similarity functions used in the proposed method.
- 25

Comment 6: Page 13, line 2, how to realize your "automatic program" to derive other attributes? Do you mean that these attributes have been processed into a dataset? Otherwise, I think it is hard to automatic match these attributes in their text manual.

Response: In this study, we manually selected the peer-reviewed papers related to the drainage network extraction applications which were published in mainstream journals of related domains. After the study area of each case was set, an automatic program was applied to SRTM DEM or ASTER GDEM of the study area to derive attributes (such as area, total relief, elevation-slope cumulative frequency distribution, and hypsometric curve) for each case. The results were recorded in the case base. We have made it clear in the revised manuscript (the second paragraph of Section 5.1.1).

12

**13 *Comment 7: Page 14, line23, 0.4->0.43.**

14 Response: We have revised the manuscript to correct it.

15

**Comment 8: Page 20, table 2, do you consider some other parameters? For example, I think the characteristics of study area are somewhat simple.**

Response: The method proposed in current study focuses on DTA domain and 18 considers the area and the terrain condition through a few simple attributes for 19 describing the study area characteristics of a DTA application case. Preliminary 20 evaluation shows the reasonableness of the proposed method. The design of the 21 attributes used to describe the problem part of a case could be improved to describe 22 the domain-specific application-context information in a more adaptive and efficient 23 manner, which needs additional research. We have revised the manuscript to discuss 24 25 this issue (the second paragraph of Section 6).

**1 Comment 9: Page 23, the overall similarity, can it be calculated using weighting?**

Response: We have tested the effect of calculating the overall similarity by a simple average operator (a representative of weighted-average) instead of the minimum operator. The experimental results show that the proposed method with a minimum operator performs more reasonablely. We have revised the manuscript to make a fruther discussion on it. Please also see our response above to the third item of comments from anonymous referee #2.

- 8
- 9

**Comment 10: Figure 4, part b. is it right?**

Response: Fig. 4b is correct. For this case, the CA threshold resulted from the proposed method is larger than it recorded in the evalution case, which means that the drainage network extracted by using the the CA threshold result is shorter than the original drainage network of this case. The situation shown in Fig. 4a is contrary. We have revised the caption of Fig. 4 to include this information.

[revised manuscript text omitted]

|---|----------------------------|
|   | mutual st. t. aba          |
|   | distribution of the CA.    |
|   |                            |

1 of the solution part of the cases is the parameter value, i.e., the CA threshold. Although this

2 experiment is somewhat simplified, we believe that it can evaluate the proposed method as

3 effectively as an experiment with a more complex design.

**4 5.1.1 Preparation of a case base**

5 The case base prepared for this experiment includes 124 cases of drainage network extraction (Fig. 3). Each case originated from a peer-reviewed article related to the target task that was 6 7 recently published in mainstream journals of related domains (such as Water Resources 8 Research, Hydrology and Earth System Sciences, Hydrological Processes, Computers & 9 Geosciences, and Advances in Water Resources; see the Appendix for the list of the articles 10 used for cases). These articles were manually selected to be as reliable as possible. They are 11 supposed to provide good solutions (might not be optimal) for their specific study areas based on experts' experience and knowledge of the target task. When a single flow direction 12 13 algorithm (such as D8 algorithm) was adopted by most of these articles (a few articles did not 14 state clearly the flow direction algorithm used), the CA threshold values adopted in these articles were highly varied (about  $10^{-3}$ – $10^{3}$  km2). 15

16 Each case was manually prepared from a journal article. The main work involved in preparing 17 the case problem was to specify each attribute of the study area, whereas the work involved in 18 preparing the case solution focused on recording the CA threshold used in the article. Normally, the cell size used is clearly stated in the article and can be filled in as the 19 20 corresponding case attribute. However, this is often not true for other attributes. Given the 21 study area of a case, an automatic program was applied to a free DEM dataset of the study area (mainly an SRTM DEM with a resolution of 90 m and an ASTER GDEM with a 22 23 resolution of 30 m) to derive the other attributes (such as area, total relief, elevation-slope 24 cumulative frequency distribution, and hypsometric curve) for each case. Original DEM 25 adopted in some articles has a finer resolution than that of SRTM DEM or ASTER GDEM. However, those DEMs are often not easy to collect. This experiment used open DEM data to 26 derive above case attributes and to make each of these attributes comparable between 27 28 different cases.

For the solution part of each case, the CA threshold given explicitly in each article was recorded directly. If the CA threshold was shown only implicitly in the drainage network figure in an article, it was determined based on visual comparison between the drainage

|----|-------------------|--|--|
| 带格 | 式的: 字体:倾斜  |  |  |
| 带格 | ·式的: 字体:倾斜 |  |  |
| 删除 | 的内容:              |  |  |

|---|---------------------|
|   |                     |


[revised manuscript text omitted]

| 骨格式的: 字体: 倾斜             |                                              |
|--------------------------|----------------------------------------------|
| 世校式的                     |                                              |
|                          |                                       |
| 市格 式的             |     (                                 |
| 帶格式的                     |                                       |
| ( 带格式的                   |                                       |
| 世校式的                     |                                              |
| (甲間八四)
(### - + #+    |                                       |
| 1 审船式的                   |                                              |
| 带格式的                     | · · · ·                                      |
| 带格式的                     | · · · ·                                      |
| 世界なりの                    |                                              |
|                          |                                       |
| (市俗 丸 的)          |                                       |
| 带格式的                     | · · · ·                                      |
| 世校式の                     |                                       |
| 中 111 人印)
**# + # #  |                                       |**
| 市哈 瓦的
**##   |                                       |**
| 市哈 式的             |                                              |

[revised manuscript text omitted]

**...**

(...

...

(...

(...

(...

**...**

(...

...

· · · ·

|---------------------|-----------------------|------|------|--------------------------|-------------|----------------|-------------------------|-----------------------|--|
| Chili-Wene [0,24]   | E.1.W. [0.22]         | 0.90 | 0.17 | E-1W- [0 22]             | 0.00        | 0.17 *         |                         | 带格式的 世故式的             |  |
| Chijiawang [0.54]   | Ernwu [0.23]          | 0.80 | 0.17 |                          | 0.89        | 0.17    | X                       | 带格式的                  |  |
| Hailogou [2.03]     | SanPabloLaMana [3.07] | 0.68 | 0.18 | HunzaRiver[56.7]         | 0.79        | 0.79    |                         | 带格式的                  |  |
| Batchawana [0.75]   | ClearCreek [1.22]     | 0.58 | 0.2  | XianNanGou[0.004]        | 0.81 | 17.16 • |                         | 带格式的                  |  |
| Liene [5.37]        | LiWuRiver [9]         | 0.74 | 0.2  | LiWuRiver [9]            | 0.85        | 0.2            | $\square$               | 带格式的                  |  |
| Zwalm [0.36]        | Haean [0.55]          | 0.73 | 0.2  | Haean [0.55]             | 0.87        | 0.2.           | $\mathbb{N}/\mathbb{N}$ | 带格式的
一带格式的         |  |
| TanaiosRiver [2720] | SaoFrancisco [5160]   | 0.67 | 0.23 | SaoFrancisco [5160]      | 0.84        | 0.23           |                         | 带格式的                  |  |
| Durdalzin [502]     | Mahanadi Diyar [201]  | 0.00 | 0.24 | Mahanadi Divar [201]     | 0.05        | 0.24           |                         | 带格式的                  |  |
| Durdekili [302]     | Mananadikivei [891]   | 0.90 | 0.24 |                          | 0.95        | 0.24           |                         | 带格式的                  |  |
| Garonne [247.68]    | PoRiver [486]         | 0.71 | 0.24 | PoRiver [486]            | 0.87        | 0.24           |                         | 带格式的                  |  |
| NorthEsk [1.22]     | SanPabloLaMana [3.07] | 0.63 | 0.33 | UpperGuadiana[324]       | 0.82        | 0.98    |                         | 带格式的                  |  |
| YbbsRiver [1.01]    | Davidson [0.48]       | 0.69 | 0.43 | CameronHighlands[0.0093] | 0.84        | 11.44          |                         | 带格式的带格式的              |  |
| Cordevole [0.68]    | SouthForkNew [2.7]    | 0.69 | 0.46 | HJA[0.27]                | 0.83        | 0.67.          |                         | 带格式的                  |  |
| NaravaniRiver [130] | Durance [51 21]       | 0.51 | 0.52 | HunzaRiver[56 7]         | 0.75        | 0.45           |                         | 带格式的                  |  |
| YaluTsangpo [81.56] | SalmonRiver [486]     | 0.47 | 0.55 | RhoneRiver[40.5]         | 0.68        | 0.41           |                         | 伊格  九的
世故式的 |  |
| Kasilian [0.08]     | Haean [0.55]          | 0.63 | 0.63 | Haean [0.55]             | 0.83        | 0.63    |                         | 带格式的                  |  |
| UpstreamGarza [0.2] | NorsmindeFjord [4.05] | 0.69 | 0.74 | Haean [0.55]             | 0.83 | 0.37    |                         | 带格式的                  |  |
| Zhanghe [33.11]     | Longuen [7.29]        | 0.69 | 1.06 | Longuen [7.29]           | 0.89        | 1.06           |                         | 带格式的                  |  |

(...

(...

(...

...

(...

...

(...

**...**

**...**

**...**

**...**

...

····

|                                    | S∈[0.8,1] | S∈[0.7,0.8) | S∈[0.6,0.7) | S∈[0,0.6) | Total count of cases |
|------------------------------------|-------------------------|--------------------|--------------------|------------------|----------------------|
| E∈[0,0.1]                   | 10               | 11          | 3           | 2         | 26            |
| E∈(0.1,0.25]                | 3                | 8           | 4           | 1         | 16            |
| E∈(0.25,0.5]                | 0                | 0           | 3           | 0         | 3             |
| $\underline{E} \in (0.5, +\infty)$ | 0                | 0           | 3           | 2         | 5             |

1 Table 4. Relationship between E and the similarity value (S) of the solution case to the 2 evaluation case.

3

**1 Table 5. Top 10 similarity values between the YbbsRiver evaluation case and existing cases**

**2 as reasoned by the proposed method.**

|                | Simil        | arity va | lue on ind                     | ividual attribute                          |                          |                    |              |                                                    |
|----------------|--------------|----------|--------------------------------|--------------------------------------------|--------------------------|--------------------|--------------|----------------------------------------------------|
| Case name      | Cell
size | Area     | Total
r elief | Elevation-
slope
distribution | Hypso
metric
curve | Overall similarity | E     | 删除的内容: Relief-slope         带格式表格         删除的内容: R |
| UpperMcKenzie  | 1            | 0.73     | 0.90                           | 0.62                                       | 0.92                     | 0.62               | 0.4 3 |                                                    |
| XianNanGou     | 0.58         | 0.61     | 0.88                           | 0.59                                       | 0.76                     | 0.58               | 21.73        |                                                    |
| NorsmindeFjord | 0.58         | 0.74     | 0.84                           | 0.64                                       | 0.91                     | 0.58               | 0.44         |                                                    |
| Pettit         | 1            | 0.56     | 0.96                           | 0.62                                       | 0.76                     | 0.56               | 1.19         |                                                    |
| Bellebeek      | 0.54         | 0.69     | 0.83                           | 0.54                                       | 0.81                     | 0.54               | 0.73         |                                                    |
| Haean          | 0.51         | 0.65     | 0.94                           | 0.78                                       | 0.93                     | 0.51               | 0.33         |                                                    |
| MicaCreek2     | 0.51         | 0.53     | 0.89                           | 0.62                                       | 0.75                     | 0.51               | 5.23         |                                                    |
| SouthForkNew   | 0.51         | 0.69     | 0.89                           | 0.76                                       | 0.52                     | 0.51               | 0.35         |                                                    |
| Babaohe        | 0.51         | 0.57     | 0.88                           | 0.73                                       | 0.90                     | 0.51               | 0.73         |                                                    |
| ClintonRiver   | 0.51         | 0.59     | 0.85                           | 0.56                                       | 0.55                     | 0.51               | 0.79         |                                                    |

**1 Table 6. Top 10 similarity values between the Kasilian evaluation case and existing cases as**

**2 reasoned by the proposed method.**

|                | Simil        | arity valu | ie on indi             | ividual attribute                          |                          | _                  |      |        |                                          |
|----------------|--------------|------------|------------------------|--------------------------------------------|--------------------------|--------------------|------|--------|------------------------------------------|
| Case name      | Cell
size | Area       | Total
relief | Elevation-
slope
distribution | Hypso
metric
curve | Overall similarity | E    | ×
| Haean          | 0.63         | 0.92       | 0.83                   | 0.83                                       | 0.93                     | 0.63               | 0.63 |        |                                          |
| SanPabloLaMana | 0.61         | 0.61       | 0.74                   | 0.60                                       | 0.76                     | 0.60               | 0.84 |        |                                          |
| Brue           | 0.61         | 0.67       | 0.73                   | 0.59                                       | 0.88                     | 0.59               | 0.66 |        |                                          |
| OitaRiver      | 0.61         | 0.57       | 0.95                   | 0.73                                       | 0.96                     | 0.57               | 0.91 |        |                                          |
| Baba           | 0.61         | 0.55       | 0.98                   | 0.83                                       | 0.97                     | 0.55               | 0.87 |        |                                          |
| JuniataRiver   | 0.63         | 0.55       | 0.78                   | 0.64                                       | 0.86                     | 0.55               | 0.92 |        |                                          |
| NorsmindeFjord | 0.54         | 0.74       | 0.71                   | 0.72                                       | 0.95                     | 0.54               | 0.87 |        |                                          |
| Lonquen        | 0.61         | 0.52       | 0.82                   | 0.73                                       | 0.93                     | 0.52               | 0.92 |        |                                          |
| HJA            | 0.63         | 0.90       | 0.86                   | 0.51                                       | 0.64                     | 0.51               | 0.48 |        |                                          |
| Bellever       | 0.61         | 0.78       | 0.74                   | 0.50                                       | 0.68                     | 0.50               | 0.63 |        |                                          |

---

## Author Response (AR2)

**Revisions and responses on HESS-2015-539 ("Case-based knowledge formalization and reasoning method for digital terrain analysis ─ Application to determining the catchment area threshold for extracting drainage networks")**

The authors thank Dr. H. Mitasova for the comments on our manuscript. We make a point-by-point reply to these comments as below. A marked-up manuscript version showing the changes made is attached at the end of this document.

*Comment 1: The revised manuscript now states clearly that when the DEM used in the case study was not available the authors used SRTM or ASTER DEM instead. It is important to show this in the Table 3 by including the resolution of the original DEM used in the articles (e.g. in the brackets along with the CA threshold) where this information is available..*

Response: Considerring that the Appendix lists all cases used in the manuscript and Table 3 in the manuscript lists 50 evaluation cases among them, we revised the Appendix to add the resolution of the DEM originallly used in every case in the table of Appendix. We also revised the manusctipt to declare that this information was included in the Appendix (the second paragraph of Section 5.1.1).

*Comment 2: Throughout the article the term "relieve" is not used in the right context (please see http://www.dictionary.com/browse/relieve?s=t) leading to misunderstandings. Perhaps you wanted to say reduce, minimize or eliminate?*

Response: Thanks for the suggestion. We have revised the manuscript to use "reduce" and "minimize" instead of the "relieve" used at several places in the original manuscript.

*Comment 3: Another case of misleading terminology is "magnitude of cell size" noted by two reviewers. The sentence where it is used "The difference in magnitude of cell size can better reflect the level of similarity between DTA 26 applications than the numerical difference in cell size." really does not make much sense. Perhaps you wanted to say that you used "absolute value of the difference in cell size"?*

Response: The difference in the logarithmic value of cell size can better reflect the level of similarity between DTA applications than the numerical difference in cell size. The greater the difference in the logarithm of cell size, the lower is the similarity. We have revised the manuscript to make it clear (the second paragraph of Section 4.2.2).

*Comment 4: One of the reviewers suggested that the title is too complex, however, in the revised paper the authors made it even longer and more complex - shorter title would be desirable.*

[revised manuscript text omitted]